# NEBULA: Do We Evaluate Vision-Language-Action Agents Correctly?

## Abstract

The evaluation of Vision-Language-Action (VLA) agents is hindered by the coarse, end-task success metric that fails to provide precise skill diagnosis or measure robustness to real-world perturbations. This challenge is amplified by scattered data that limits reproducibility and progress toward generalist models. To address these limitations, we introduce **NEBULA**, a unified ecosystem for single-arm manipulation that enables diagnostic and reproducible evaluation. NEBULA features a novel dual-axis evaluation protocol that combines fine-grained *capability tests* for precise skill diagnosis with systematic *stress tests* that measure robustness. A standardized API and a large-scale, aggregated dataset are provided to reduce fragmentation and support cross-dataset training and fair comparison. Using NEBULA, we demonstrate that top-performing VLAs struggle with key capabilities such as spatial reasoning and dynamic adaptation, which are consistently obscured by conventional end-task success metrics. By measuring both what an agent can do and when it does so reliably, NEBULA provides a practical foundation for robust, general-purpose embodied agents.

## 1 Introduction

Vision–Language–Action (VLA) agents are advancing rapidly, spanning language-conditioned planners, generalist multi-modal agents, and prompt-conditioned manipulation policies (Brohan et al., 2023; Zitkovich et al., 2023; Jiang et al., 2022). Yet a basic question remains: *are we evaluating what actually matters?* Most benchmarks tend to prioritize end-task success, a coarse metric that neither reveals which subskills are engaged nor localizes error sources. For example, a failure on "pick-and-place" may arise from language grounding, 3D perception, spatial planning, or control. However, a single success rate cannot identify the failing component. Without capability-resolved, diagnostic evaluation, we cannot measure per-skill capability and expose where and why agents fail.

Even with precise skill diagnosis, current evaluation overlooks a second deployment-critical dimension: reliability. Passing a test at a single operating point does not imply robustness, nor does it reflect key properties needed for deployment (*e.g.,* latency, stability, robustness). Small, realistic shifts in conditions (*e.g.,* lighting, textures, phrasing, dynamics, sensor noise) can flip outcomes, while aggregate success rates often hide variability across settings and mask abrupt breakdowns ('failure cliffs'). Because real-world conditions continually shift along these dimensions, stress tests are needed to characterize reliability boundaries and disentangle competence from robustness.

Meanwhile, this dual challenge of diagnostic and robust evaluation is compounded by a severely fragmented data landscape. Datasets like ManiSkill (Mu et al., 2021), LeRobot (Cadene et al., 2024), and BEHAVIOR-1k (Li et al., 2023) differ drastically in format, task representation, and embodiment. Even efforts like Open-X (Collaboration et al., 2023), which propose shared interfaces, fall short in defining what capabilities are tested or how to compare them. This fragmentation forces researchers to reimplement pipelines, prevents fair head-to-head comparisons, and limits large-scale generalization studies, which slows progress toward unified embodied intelligence. As a result, the field lacks a unified ecosystem that can simultaneously diagnose agent capabilities, stress-test their robustness, and unify disparate data sources for reproducible, scalable research.

To address these challenges, we introduce **NEBULA**, an integrated ecosystem designed to shift the focus of embodied AI research from simple task completion to true capability mastery. The ecosystem is built on two core pillars. The first is a **diagnostic evaluation framework** that transforms

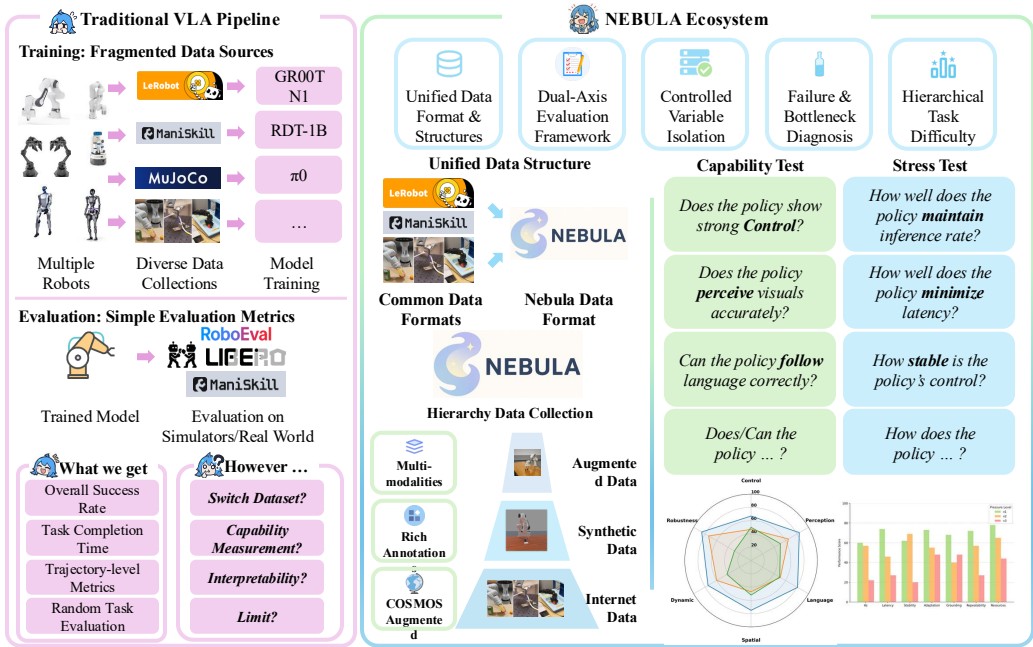

Figure 1: **NEBULA Ecosystem** unifies fragmented VLA datasets and APIs for cross-dataset training and benchmarking. It introduces a dual-axis evaluation (capability and stress testing) with controlled variable isolation for skill-specific diagnosis. With hierarchical task difficulty, multi-modal annotations, and visual performance summaries, NEBULA converts success rate into a diagnostic signal, exposing failure modes and reliability limits.

coarse success rates into an interpretable, multi-faceted signal. It directly confronts the issues of disentangled capability evaluation and reliability by combining: (1) **Capability Tests**, which isolate specific skills like spatial reasoning and grasp synthesis to pinpoint precise reasons for failure, and (2) **Stress Tests**, which systematically vary environmental conditions to map an agent's robustness and identify hidden failure cliffs. This dual-axis approach provides a holistic view of an agent's competence, revealing not only what it can do but also the conditions under which it can be trusted.

Complementing its evaluation suite, NEBULA's second pillar tackles the critical issue of **data and tooling fragmentation**. We provide a standardized API and data format that unifies disparate benchmarks, including ManiSkill, LeRobot, and others, eliminating the need for engineering work for each new dataset. We also provide a large-scale, aggregated dataset that integrates existing real-world demonstrations, simulator-generated trajectories, and world-model–augmented data. By providing the infrastructure for both unified training and reproducible evaluation, NEBULA empowers researchers to build more generalizable agents and conduct fair, large-scale comparisons that accelerate scientific progress. In summary, the key contributions of our paper include:

- We introduce **NEBULA**, a unified VLA ecosystem that provides a standardized API and a large-scale, aggregated dataset to facilitate reproducible, cross-dataset training and benchmarking.

- We propose a novel **dual-axis evaluation protocol** that combines fine-grained *capability tests* for precise skill diagnosis with systematic *stress tests* to measure an agent's robustness against real-world perturbations.

- We present an **in-depth benchmarking study** of current VLAs, revealing critical failure modes (*e.g.,* spatial reasoning) that are typically obscured by the traditional success rate metric.

## 2 RELATED WORKS

### 2.1 SINGLE-ARM MANIPULATION BENCHMARKS & SIMULATORS

The landscape of robotic manipulation evaluation has expanded significantly in recent years, yet fundamental questions about what and how we measure remain unresolved. Existing efforts cluster

into three threads: (i) **Single-arm tabletop benchmarks**, such as RLBench (James et al., 2020), BulletArm (Wang et al., 2022), ManiSkill2 (Gu et al., 2023), and ManiSkill3 (Tao et al., 2024), provide diverse task libraries, multimodal observations, and extensions toward bimanual, language-conditioned manipulation. (ii) **Long-horizon benchmarks**, such as BEHAVIOR-1K (Li et al., 2023), Meta-World (Yu et al., 2020), ALFRED (Shridhar et al., 2020), FurnitureBench (Heo et al., 2023), Franka Kitchen (Gupta et al., 2019), LIBERO (Liu et al., 2023), CALVIN (Mees et al., 2022), VLABench (Zhang et al., 2024), and MIKASA-Robo (Cherepanov et al., 2025), highlight multi-skill acquisition, temporal reasoning across extended tasks, and memory-centric challenges under partial observability. (ii) **Realism-focused platforms**, such as SIMPLER (Li et al., 2024), Habitat (Savva et al., 2019), SAPIEN (Xiang et al., 2020), THE COLOSSEUM (Pumacay et al., 2024), and Genesis (Authors, 2024), advance physics fidelity and enable evaluation under controlled perturbations and language-conditioned tasks.

Despite these advances, evaluation in most benchmarks still relies heavily on task-level success rate. While useful for model comparison and easy to compute, these metrics have limited diagnostic value: they indicate neither which abilities failed nor why. Our framework addresses this gap through a dual-axis evaluation that disentangles task requirements from performance quality, enabling structured and interpretable diagnosis.

## 2.2 EVALUATION PROTOCOLS & METRICS

Separating sources of failure is essential for evaluating VLA models in robotic manipulation, particularly as tasks grow more complex and as the demand for stronger generalization increases. THE COLOSSEUM (Pumacay et al., 2024) systematically perturbs tasks along controlled axes and reports robustness degradation. VLABench (Zhang et al., 2024) divides evaluation into six high-level capability dimensions to assess models more explicitly. RAMP (Robotic Assembly Manipulation and Planning) (Collins et al., 2023) introduces long-horizon assembly scenarios that challenge reasoning, diagnostics, and fault recovery in addition to pure control and perception. Meanwhile, Robot Policy Evaluation for Sim-to-Real Transfer (Yang et al., 2025) proposes benchmarking strategies that gradually increase task complexity and introduce scenario perturbations to assess robustness and alignment between simulation and real-world performance. Also, Recent surveys on VLA models emphasize the need for evaluation across the full perception–language–control pipeline, combining task success with metrics for generalization, robustness, and instruction understanding (Ma et al., 2024; Shao et al., 2025; Sapkota et al., 2025). Our work builds on the idea of evaluating intelligence across multiple dimensions, and further enforces protocols that disentangle task specifications from execution performance via controlled variation and progressively increasing difficulty.

## 3 NEBULA ECOSYSTEM

NEBULA is a unified and comprehensive ecosystem built to overcome critical limitations in existing Embodied AI pipelines. While traditional systems often reduce evaluation to coarse metrics like task success or runtime, NEBULA broadens the scope to answer a deeper question: *how and why does an agent succeed or fail?* As shown in Figure 1, NEBULA provides a structured, modular framework that includes 1) a standardized data layer with a unified format and APIs to enable cross-task training and reuse; 2) a dual-axis evaluation protocol for disentangling functional capabilities from real-time robustness; and 3) rich diagnostic outputs to support interpretable, skill-specific performance analysis. This section introduces NEBULA's core design. Section 3.1 details our data collection protocol and unified API design, while Section 3.2 outlines NEBULA's evaluation framework and the design of its capability and stress test tasks.

### 3.1 DATA & API SPECIFICATION

**Data Collection & Annotation.** To ensure consistency and reproducibility, NEBULA collects all training and evaluation data using a customized simulation platform built upon the SAPIEN (Xiang et al., 2020) engine and the ManiSkill3 (Tao et al., 2024) framework. For each manipulation episode, we record a temporally ordered sequence of multimodal observations $\mathcal{O}_t$, system states $\mathcal{S}_t$, actions $\mathcal{A}_t$, and binary success labels $\mathcal{SU}_t \in 0, 1$ at each timestep $t$. The observations $\mathcal{O}_t$ include RGB, depth, and segmentation images from six fixed-viewpoint cameras, as well as proprioceptive inputs such as joint positions $q_t$ and velocities $\tilde{q}_t$. Each episode is annotated with a natural lan-

guage task instruction, manually written to reflect the intended goal. These instructions serve as the conditioning input for language-conditioned policies and allow precise alignment between episodes and their semantic objectives. NEBULA offers two dataset variants, Alpha and Beta, designed to balance completeness and usability. For data collection, the Alpha version of the dataset is entirely generated using expert trajectories produced via motion planning (LaValle, 2006). In contrast, the Beta version combines motion planning with human teleoperation: for selected hard tasks, expert demonstrations are collected manually to capture more diverse and realistic behaviors.

**API & Modulated Assets.** To ensure consistency and ease of use across heterogeneous data sources, we introduce a unified data schema that consolidates fields found in modern embodied datasets. This schema standardizes the representation of observations, actions, environment states, and task meta-data under a common structure, enabling plug-and-play compatibility with a wide range of learning algorithms. We provide a PyTorch API that abstracts away the low-level data loading and indexing details, exposing a clean, task-agnostic interface for pipeline. For researchers working in the TensorFlow ecosystem, we additionally provide lightweight TF-compatible adapters. To further reduce integration overhead, we include model-specific adapters for several widely used architectures, allowing for immediate benchmarking on NEBULA data with minimal code changes.

**Dataset Statistics.** NEBULA offers two dataset variants—Alpha and Beta—for both full-scale evaluation and lightweight experimentation. As shown in Table 1, Alpha includes over 54,000 expert demonstrations across five capability families, while Beta is a compact version ( 10% per task) designed for rapid development and ablation. Some high-difficulty Beta tasks use human teleoperation to introduce realistic variations. Both datasets provide multimodal inputs (videos, language, trajectories) in PyTorch and TFRecord formats with adapter support. The Robustness and Generalization family is reserved for evaluation only and excluded from both training

Table 1: Dataset statistics of NEBULA-Alpha across five task families, excluding Robustness.

| Task Families | Alpha | | |
| --- | --- | --- | --- |
| | Videos | Descriptions | Traj |
| Control | 54,000 | 9 | 36,000 |
| Perception | 54,000 | 9,000 | 36,000 |
| Language | 48,000 | 24,000 | 96,000 |
| Dynamic | 36,000 | 6 | 24,000 |
| Spatial | 30,000 | 5,000 | 24,000 |
| Robust | N/A | N/A | N/A |
| Total | 222,000 | 38,015 | 216,000 |

sets to prevent overfitting and ensure fair comparison under distribution shift.

### 3.2 DUAL-AXIS EVALUATION FRAMEWORK

NEBULA introduces a dual-axis evaluation framework to enable structured, interpretable, and diagnostic assessment of embodied AI systems. This framework decouples the evaluation into two dimensions: **Capability** and **Stress Tests**, each isolating a distinct facet of system performance. The Capability axis evaluates *what the agent can do* under nominal conditions. The Stress axis probes *how well the agent operates* under varying levels of real-time or robustness-related pressure.

#### 3.2.1 CAPABILITY TEST TASKS

NEBULA isolates six core functional capabilities, distinct from the temporal sub-tasks (*e.g.,* reach, grasp) used in traditional benchmarks. While sub-tasks identify *where* a failure occurs, they often fail to explain *why*. For instance, a grasp failure may stem from perceptual, reasoning, or control errors. To address this, our methodology prioritizes functional decomposition through two key principles: *(i) Controlled-Variable Isolation*: We vary a single capability dimension while holding others constant (*e.g.,* minimizing control complexity to test perception), ensuring performance changes are directly attributable to the target skill. *(ii) Systematic Difficulty Scaling*: Tasks are generated via parameterized templates into three tiers (Easy, Medium, Hard) to provide a fine-grained analysis of an agent's limits. This structure enables reproducible evaluation of the following capabilities (see Fig. 2 and Appendix A.1):

(1) **Control**: The Control task family isolates low-level manipulation by fixing non-control factors, with tasks progressing from simple actions (Easy) to precise, multi-step sequences (Hard).

(2) **Perception**: The Perception task family isolates visual recognition by minimizing control demands, with difficulty scaling from clear distinctions to subtle differences and cluttered scenes.

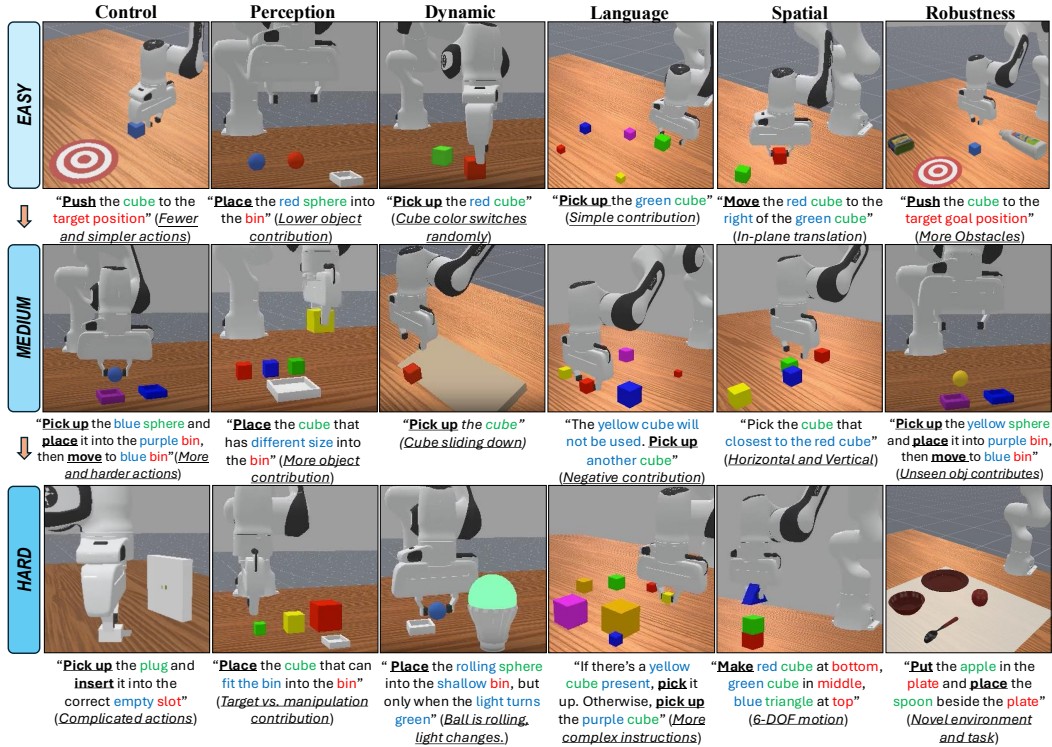

Figure 2: **Examples of NEBULA Capability Test task** across six core capabilities (Control, Perception, Dynamic Adaptation, Language, Spatial Reasoning, and Robustness) organized into three difficulty levels. Tasks isolate specific skills with controlled complexity. Green marks objects, red marks targets, and blue indicates contextual cues. **Bold underlined** text shows actions; *italic underlined* text gives clarifications.

(3) *Dynamic Adaptation*: This task family evaluates how well an agent adapts to dynamic changes, from object attribute switching (Easy) to predictable moving (Medium) and unpredictable real-time events (Hard), testing reactivity and robustness.

(4) *Language*: The Language task family tests instruction understanding, from basic grounding to reasoning and conditionals, with fixed scenes to isolate linguistic skills.

(5) *Spatial Reasoning*: The Spatial task family tests spatial reasoning from 2D placement to 6-DoF planning, with difficulty scaling from planar to full 3D geometric understanding.

(6) *Robustness/Generalization*: The Robustness task family assesses generalization under distribution shifts, from distractors to unseen attributes to novel scenes.

### 3.2.2 STRESS TEST TASKS

To complement capability-based evaluations, NEBULA introduces a suite of Stress Tests (*Inference Frequency*, *Stability Score*, *Adaptation*, *Color Shift*, *Noise*, *Resolution*, *Light Flicker*, *Rolling Shutter*, and *Actuator Latency*) that isolate and quantify system performance under targeted operational constraints. Each test is a single-indicator probe. These tests avoid confounding variables and support controlled ablation studies by being applied independently. Each is instantiated at three calibrated pressure levels ($v_1$–$v_3$), defined by measurable parameters. This structure enables detailed stress-response profiling and fair comparisons across systems, helping to identify bottlenecks and guide robustness optimization for real-world deployment. Fig 3 illustrates representative examples of our stress tests. Complete test descriptions are provided in Appendix A.2..

(1) **Cognitive Stressors**

    (a) *Inference Frequency*: Measures responsiveness by increasing motion complexity to reveal inference rate limitations under real-time pressure.

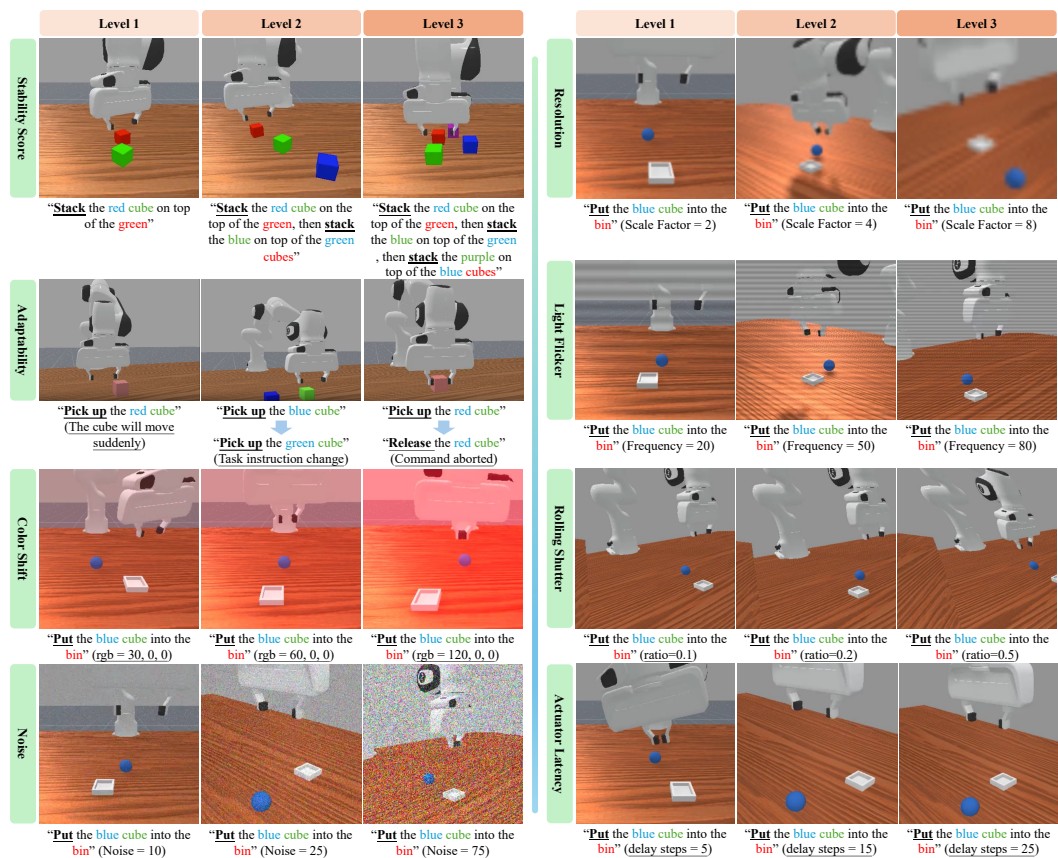

Figure 3: **Examples of NEBULA Stress Test tasks** designed to evaluate robustness under systematically varied perturbations. Each stressor (*e.g.,* object color shifts, sensor noise, control latency, lighting changes, and visual occlusion) is independently scaled across three pressure levels (v1–v3), enabling targeted stress evaluation. Green marks denote objects, red indicates targets, and blue highlights context or distractors. *Italic underlined* text clarifies dynamic changes.

  (b) *Adaptability*: Evaluates agents' ability to adjust to dynamic targets or instruction switches, testing robustness to online re-planning.

(2) **Perceptual Degradation Stressors**

  (a) *Color Shift*: Applies global RGB shifts (*e.g.,* +30,0,0) to simulate color distortion due to lighting or camera tuning.

  (b) *Noise*: Adds Gaussian and Poisson pixel-level noise to simulate sensor degradation such as low-light or transmission artifacts.

  (c) *Resolution*: Downsamples visual input to mimic low-fidelity perception (*e.g.,* poor camera quality or compression).

  (d) *Frame Drop*: Randomly drops frames to assess resilience to temporal perception gaps.

  (e) *Light Flicker*: Introduces frame-wise brightness fluctuation, emulating unstable illumination in real-world scenes.

  (f) *Rolling Shutter*: Applies line-wise distortions to simulate geometric warping during fast motion with rolling shutter cameras.

(3) **Actuation-Level Disruptions**

  (a) *Actuator Latency*: Adds fixed delays between issued commands and their execution to stress temporal coordination.

  (b) *Command Packet Loss*: Randomly drops control signals to simulate unreliable planner-actuator communication links.

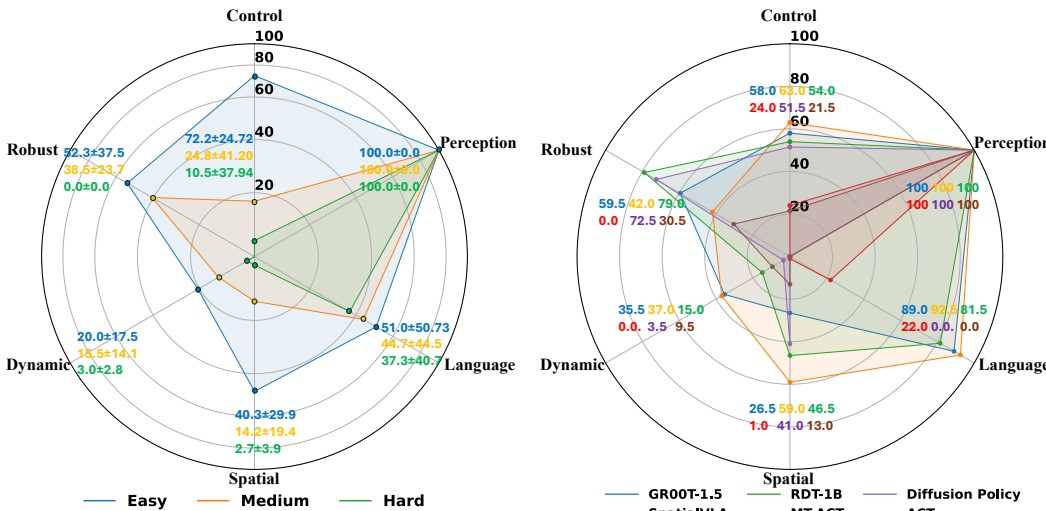

Figure 4: This figure presents two radar charts summarizing model performance across six capability task families. The **left** chart shows the mean ± standard deviation of success rates across all models for each task family at three difficulty levels. The **right** chart displays the average performance of individual models on Easy and Medium tasks, allowing for comparison across architectures.

(c) *Stability Score*: Quantifies action smoothness given a sequence $\{a_0, a_1, ..., a_t\}$ using:

$$\text{Stability} = \exp\left(-\frac{1}{T-1}\sum_{t=1}^{T}\|\mathbf{a}_t - \mathbf{a}_{t-1}\|_2\right) \tag{1}$$

Higher scores ($\in [0, 1]$) indicate smoother control trajectories. Stress levels escalate from coarse-force to contact-rich settings.

## 4 EXPERIMENTS

To demonstrate the utility, coverage, and diagnostic strength of the NEBULA benchmark, we conduct comprehensive experiments across both capability and stress axes. These evaluations aim to answer several core questions: *Can current embodied agents handle a wide range of skills? Where do they fail under specific challenges?* And *how can structured benchmarks help improve their design?* All experiments are conducted using the Alpha dataset to ensure consistency and reproducibility across tasks and conditions. This section focuses on the evaluated models (Section 4.1) and their performance under the dual-axis evaluation framework (Section 4.2 and Section 4.3).

### 4.1 BASELINES

We evaluate a diverse set of state-of-the-art embodied agents to benchmark performance across NEBULA's evaluation framework. Specifically, we include GR00T-1.5 (Bjorck et al., 2025), SpatialVLA (Qu et al., 2025), RDT-1B (Liu et al., 2024), MT-ACT (Bharadhwaj et al., 2024), Diffusion Policy (Chi et al., 2023), and ACT (Zhao et al., 2023), which together represent a wide spectrum of architectural designs and control paradigms. For fair comparison, we unify data loading to match NEBULA's format, keeping each model's architecture, loss, and hyperparameters unchanged. All models are fine-tuned on NEBULA Alpha using their original training protocols. For detailed paramaters, please refer to Appendix A.4.

### 4.2 CAPABILITY TEST RESULTS

As shown in the left chart from Figure 4, the radar chart reveals several key trends in agent capabilities. Most models demonstrate strong performance in *Perception* and *Language* tasks. Nearly all baselines reliably identify object attributes like color and shape, even with distractors, indicating robust visual recognition. Similarly, these agents exhibit solid instruction grounding, successfully

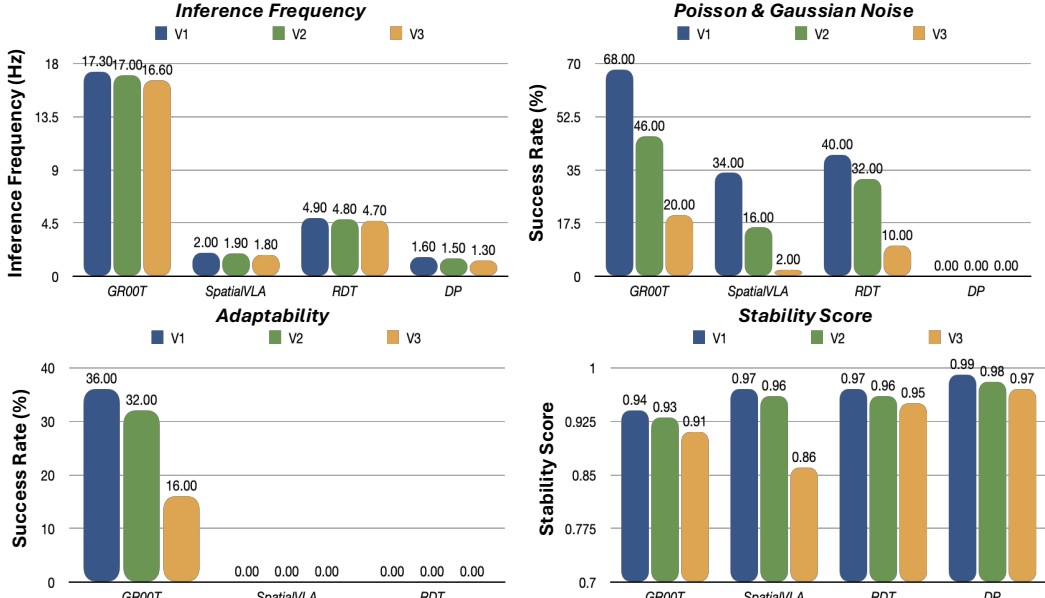

Figure 5: **Stress Test Evaluations.** Representative examples of NEBULA stress tests comparing four models under three stress levels ($v1$, $v2$, $v3$). Metrics include inference frequency (Hz), stability score (0–1), adaptability, and performance under Poisson + Gaussian Noise. Higher values indicate better performance on all metrics.

executing complex, conditional, and multi-step commands. ACT and DP do not support language-conditioned execution and thus cannot be evaluated on *Language* tasks. For completeness and comparability, we include them in the radar chart with their *Language* score set to zero by design to reflect the absence of this input modality. Interestingly, this lack of language conditioning correlates with lower performance on Spatial tasks (Fig 4), a phenomenon we analyze further in Sec 5.1.

Performance on *Control* and *Spatial* tasks is more varied. SpatialVLA and GR00T-1.5 lead in *Control*, handling long-horizon action sequences with high success. However, models like MT-ACT and ACT lag behind, revealing a need for better motor planning modules. *Spatial reasoning* remains a key bottleneck for most models, with only SpatialVLA and RDT-1B achieving moderate success. Notably, even these models show clear drops from easy to medium level, especially under occlusion and containment conditions, indicating significant room for improvement in geometric reasoning.

All models struggle on *Dynamic Adaptation* and *Robustness* tasks, as shown in Figure 4. None of the evaluated VLA systems reliably adapts to time-sensitive triggers, distractors, or goal shifts, with radar scores near zero across the board. Similarly, robustness to distribution shifts(*e.g.,* novel object appearances, unseen layouts) is consistently poor, especially at higher difficulty levels. These results expose a major gap in current VLA capabilities and highlight two urgent research directions: real-time adaptive planning and out-of-distribution generalization.

We exclude the Hard level from the right radar chart in Figure 4 because most models exhibit near-zero performance at this difficulty tier, especially in tasks involving robustness and dynamic adaptation. By focusing on Easy and Medium tasks, the chart provides clearer insights into current model capabilities. We hope this visualization encourages future research to close the gap at the Hard level, ultimately enabling more capable and resilient VLA systems.

## 4.3 STRESS TEST RESULTS

Due to space limitations, we present only representative stress test results here. For comprehensive stress test results, please refer to Appendix A.2.4.

As shown in Figure 5, all evaluated models exhibit consistent performance degradation as stress levels increase, revealing sensitivity to deployment-time challenges such as computational bottlenecks and real-time demands. Inference frequency shows a clear decline across models: GR00T-1.5B remains the most resilient, maintaining $17Hz$ under $v3$, while DP and SpatialVLA fall below $2Hz$.

This suggests many models struggle to meet real-time requirements under stress, and performance in ideal conditions may not generalize to practical deployment.

In parallel, when subjected to visual perturbations such as Poisson and Gaussian noise added directly to RGB inputs, model robustness diverges sharply. As shown in Figure 5, GR00T-1.5B maintains moderate resilience, with success rates declining from 68% under $v1$ noise to 46% ($v2$) and 20% ($v3$). In stark contrast, all other models (RDT-1B, DP, and SpatialVLA) fail completely across all levels, achieving 0% success. This indicates that vision-conditioned policies, especially those relying heavily on pixel-level embeddings, may suffer catastrophic degradation under real-world sensor noise. The pronounced performance gap also highlights GR00T's stronger visual generalization capabilities, and further supports the importance of stress testing for assessing sim-to-real reliability.

The stability score, measuring the smoothness of action trajectories (1.0 indicates perfect stability), also reveals growing fragility under stress. While RDT and DP maintain high scores above 0.95, SpatialVLA drops from 0.96 to 0.86 under $v3$, suggesting vulnerabilities in policy determinism. This decline may stem from increased decision-making stochasticity or sensitivity to input noise, leading to unreliable behaviors in dynamic scenarios where smooth, precise motion is essential.

Finally, the Adaptability results demonstrate that most current models are unable to handle dynamically evolving conditions. Among the models with language interfaces, only GR00T shows modest success under adaptive task settings, while the others fail almost completely, with near-zero success rates across stress levels. As mentioned earlier, our Adaptability axis explicitly includes mid-episode language instruction updates (*e.g.,* on-the-fly goal or constraint changes); because Diffusion Policy does not accept language inputs, we report DP as N/A on this axis. Taken together, these findings indicate a fundamental limitation in current VLA systems when faced with shifting goals, interactive feedback, or rapid environmental changes.

## 4.4 VALIDITY OF FACTOR ISOLATION

We validate NEBULA's factor isolation by comparing perception tasks in isolated vs. entangled settings. In isolation, the robot only needs to touch the correct object using simple language; in contrast, the entangled baseline requires full grasp-and-place execution involving multiple skills. As shown in Table 2, GR00T-1.5B achieves 100% success in isolated settings, but drops to 92%, 68%, and 76% when entangled. Video review shows failures are due to control and 3D spatial reasoning errors, highlighting how unrelated bottlenecks can obscure

Table 2: Success rates of GR00T-1.5 on three Perception (Easy level) tasks, comparing settings with isolated factors versus unisolated scenes with additional distractors. Results highlight the impact of scene confounding on perceptual accuracy.

| | Perception (Easy Level) | | |
|---|---|---|---|
| Isolated Factors | PlaceBigger Sphere | Place Red Sphere | Place Sphere |
| ✓ | 100 | 100 | 100 |
| ✗ | 92 | 68 | 76 |

perception performance and validating NEBULA's controlled-variable design. For comprehensive results across all tested configurations, including tasks where additional factors are deliberately introduced, please refer to Appendix A.1.3.

## 5 DISCUSSION

### 5.1 DISENTANGLING THE BOTTLENECK

NEBULA's dual-axis evaluation reveals a distinct capability profile that aggregate success rates obscure. Under nominal conditions, models consistently demonstrate strong semantic perception and language grounding, yet exhibit substantial variability in control, spatial reasoning, and dynamic adaptation. This discrepancy suggests that the community has largely solved the problem of static recognition and instruction parsing; the critical bottleneck has shifted to the "Action Head". Specifically, the ability to translate high-level semantic plans into precise, physically grounded low-level execution.

However, our Stress Tests expose a critical fragility in this perceptual competence. While semantic recognition is strong in clean environments, sensor-level robustness remains a major failure mode. Most architectures suffer catastrophic degradation under real-world signal perturbations—such as

image noise, rolling shutter, or lighting shifts—indicating that their visual representations lack the invariance required for physical deployment. When combined with the "Real-Time Barrier" of low inference frequency, these findings highlight that future generalist agents must prioritize signal robustness and execution latency alongside reasoning capabilities to ensure safe, reliable operation.

## 5.2 WHY ARE GENERALIZATION AND DYNAMIC PERFORMANCE POOR?

To investigate why embodied agents struggle with generalization and dynamics, we decoupled the vision-language backbone from the action head. Using SpatialVLA and GR00T's VLMs, we prompted high-level plans from static NEBULA scenes and had human annotators assess their validity.

As shown in Table 3, the standalone VLMs produce consistently valid strategies even in robustness tasks. However, their integrated VLA counterparts fail to execute these plans, with success rates dropping to zero even with moderate difficulty. This highlights a critical bottleneck: strong reasoning from VLMs does not guarantee successful embodied behavior, due to limitations in the action heads' ability to translate abstract plans into precise control actions.

Table 3: Comparison of standalone VLM planners (PaliGemma, Qwen) vs. End-to-End VLA policies (SpatialVLA, GR00T) on robustness tasks

| Model | Robust/Generalization | |
|---|---|---|
| | Easy StackCube | Medium StackCube |
| PaliGemma | 85 | 75 |
| SpatialVLA | 0 | 0 |
| Qwen | 100 | 90 |
| GR00T-1.5 | 75 | 0 |

This issue is compounded by the inadequacy of conventional benchmarks that rely solely on success rate, obscuring whether failures arise from perception, reasoning, or control. NEBULA's dual-axis evaluation addresses this by disentangling high-level reasoning from low-level execution and revealing weaknesses under stress, offering the diagnostic granularity needed to build more robust and generalizable embodied systems.

## 5.3 FAST INFERENCE VIA DUAL-SYSTEM ARCHITECTURES

NEBULA's Dynamic Adaptation results (Figure 4) and Adaptation Stress Tests (Table 4) expose a fundamental Real-Time Barrier in monolithic VLA architectures. While models like RDT-1B demonstrate strong semantic planning, they fail catastrophically in dynamic environments (0% adaptability success) due to low inference frequencies ($< 2$ Hz) and high latencies ($> 500$ ms) 1. This evidence suggests that the autoregressive generation process is inherently too slow for the high-bandwidth control loops required for reactive interaction.

Table 4: Comparison of inference speed, latency, and adaptation score across models. Only GR00T-1.5 demonstrates meaningful adaptation, likely due to its significantly lower latency and higher inference frequency.

| Model | Avg. Inference Frequency | Avg. Latency | Avg. Adaptation |
|---|---|---|---|
| GR00T-1.5 | 16.98 Hz | 58.62 ms | 28 |
| RDT-1B | 4.84 Hz | 206.77 ms | 1 |
| SpatialVLA | 1.92 Hz | 520.48 ms | 0 |

In sharp contrast, GR00T-1.5 is the only model that demonstrates meaningful adaptability (28% success). This performance correlates directly with its superior responsiveness (16.98 Hz inference frequency and 58.62 ms latency). Crucially, this speed is achieved through a hierarchical "System 1 / System 2" architecture. By decoupling a slow, deliberative VLM planner from a high-frequency diffusion "spinal cord," GR00T bridges the gap between reasoning and reflexes. These findings indicate that the Real-Time Barrier is not merely a data scaling issue but a structural one; future generalist agents must likely follow this dual-system paradigm, treating high-frequency control and low-frequency reasoning as distinct, albeit coupled, computational problems.

## 6 CONCLUSION

In this work, we introduced NEBULA, an evaluation-first ecosystem that unifies fragmented data formats and establishes a dual-axis framework for embodied AI. By disentangling capability tests from stress tests and enforcing controlled-variable isolation, NEBULA provides interpretable, skill-specific, and robust performance diagnostics that go beyond conventional success rates. Our experiments demonstrate how this design reveals hidden bottlenecks, clarifies model strengths and weaknesses, and lays the foundation for systematic progress toward reliable VLA agents.

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

# A  APPENDIX

## A.1  CAPABILITY TESTS

### A.1.1  DESIGN PHILOSOPHY

Capabilities in NEBULA are defined as operational abstractions rather than theoretical cognitive modules or task-subtask hierarchies. Their purpose is to provide factor-isolated evaluation axes that reveal why an agent succeeds or fails, independent of the surface structure of a task. Each capability test is designed so that a single skill becomes the dominant bottleneck, enabling interpretable and consistent attribution across diverse agents and settings.

Crucially, NEBULA does not assume that these capabilities are perfectly independent. In real manipulation, skills such as control and spatial reasoning naturally interact. Instead of enforcing strict orthogonality, our design adopts a controlled-isolation philosophy: for each capability, we systematically suppress or hold constant other contributing factors (e.g., simplifying motion when testing spatial reasoning, fixing spatial layout when testing control precision). This approach follows two core principles:

- *Variable Control*: minimizing confounding factors so that one capability governs task difficulty
- *Capability Dominance*: tasks are constructed such that improving the targeted skill measurably affects performance, while unrelated skills have minimal influence.

This pragmatic isolation strategy enables NEBULA to support interpretable bottleneck analysis, without claiming that the six capabilities form an exhaustive or mutually exclusive taxonomy.

### A.1.2  CAPABILITY DEFINITIONS & SEPARATION

This section provides the full specification for all capability tasks used in NEBULA. Each task family targets a specific embodied competency and includes *Easy*, *Medium*, and *Hard* tiers. Each task tier comprises three unique task sets instantiated from templates, with randomized object attributes and layout to ensure diversity. Success is determined through well-defined, programmatic criteria based on object positions, interactions, and task logic.

**Control**  Control refers to an agent's ability to execute precise and coordinated low-level actions. This includes fine-grained motion planning, trajectory execution, and motor stability across diverse object configurations and constraints. The observable signal of control competence lies in smooth, purposeful action trajectories that bring manipulated objects to their target spatial arrangements without deviation or instability. Representative visualizations and their corresponding task commands are provided in Figure 6.

To isolate control capabilities, NEBULA's task design deliberately removes all perceptual, semantic, and linguistic confounders. Each task instance provides fixed object positions, orientations, and visual attributes, and the language command is deterministic and non-ambiguous. This ensures that success depends solely on motor-level execution, uninfluenced by higher-level reasoning or perception.

While control and spatial reasoning are often interdependent, the Control task family minimizes reliance on external scene understanding. Success is not determined by planning over unknown spaces but by producing stable and accurate action sequences in known, fully specified environments. In contrast to Spatial Reasoning, which involves reasoning over object-object relations and partial observability, Control assumes full information and tests the fidelity of execution alone.

An example Control task involves pushing a fixed-color cylinder from one known location to another, with success determined by precise goal alignment. The *Easy* tier might require a single push, while the *Hard* tier may involve multi-object rearrangement with sub-centimeter tolerance. Agents that lack strong control skills often exhibit failure patterns such as jittery motions, object displacement errors, or breakdowns during fine manipulation sequences.

**Perception**  The Perception capability targets the agent's ability to identify and distinguish object-level visual attributes, such as color, shape, and size, under varying levels of visual complexity.

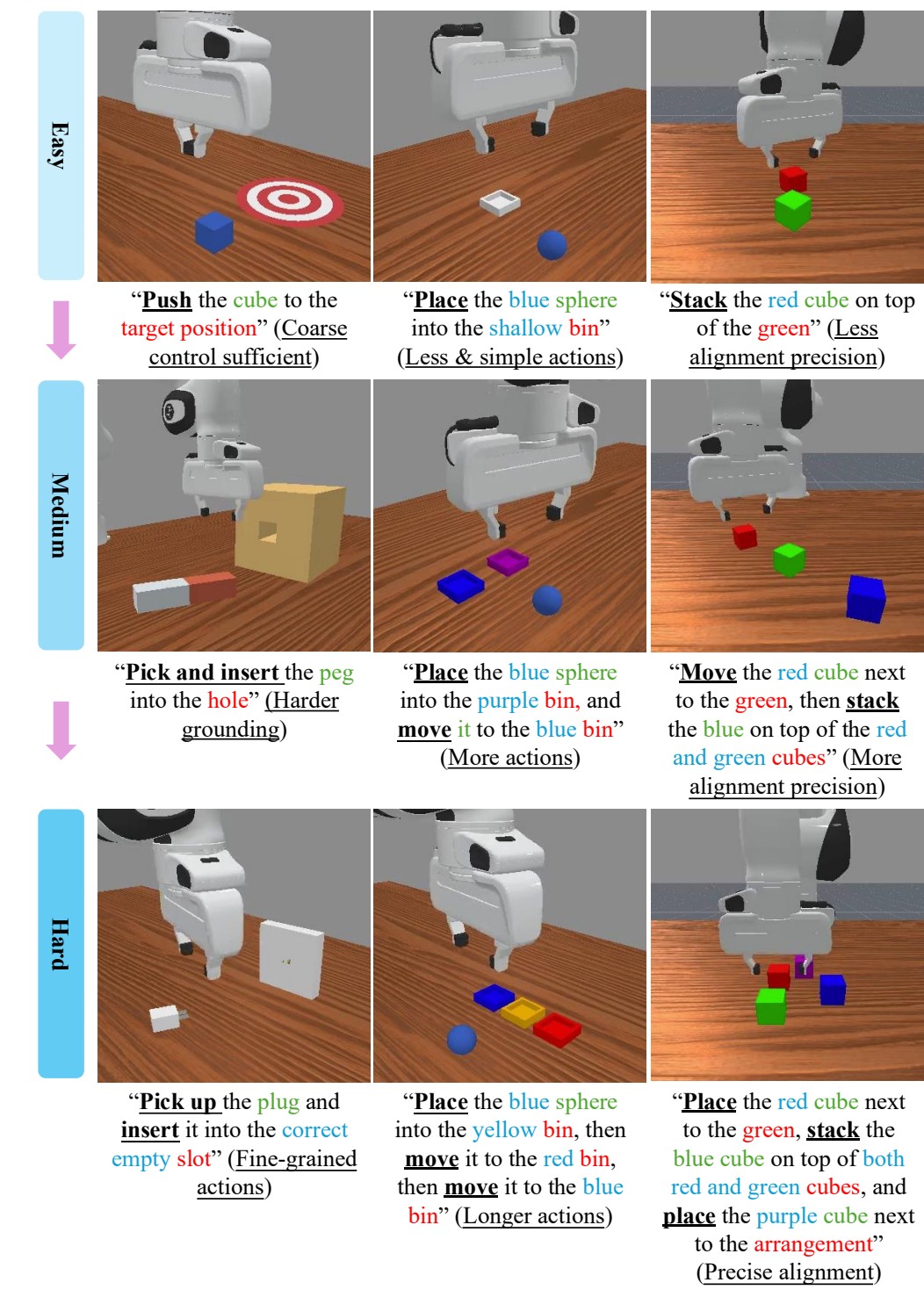

Figure 6: Visualization of NEBULA *Control* tasks. Each example highlights low-level manipulation skills while minimizing perceptual or semantic confounds. Objects are shown in green, targets in red, and contextual cues in blue. **Bold underlined** text marks required actions. Tasks span *Easy*, *Medium*, and *Hard* tiers, progressing from atomic actions to long-horizon, precision-based control.

The core skill under evaluation is visual discrimination, independent of semantic understanding or motor execution. The primary observable signal is whether the agent interacts with the correct object

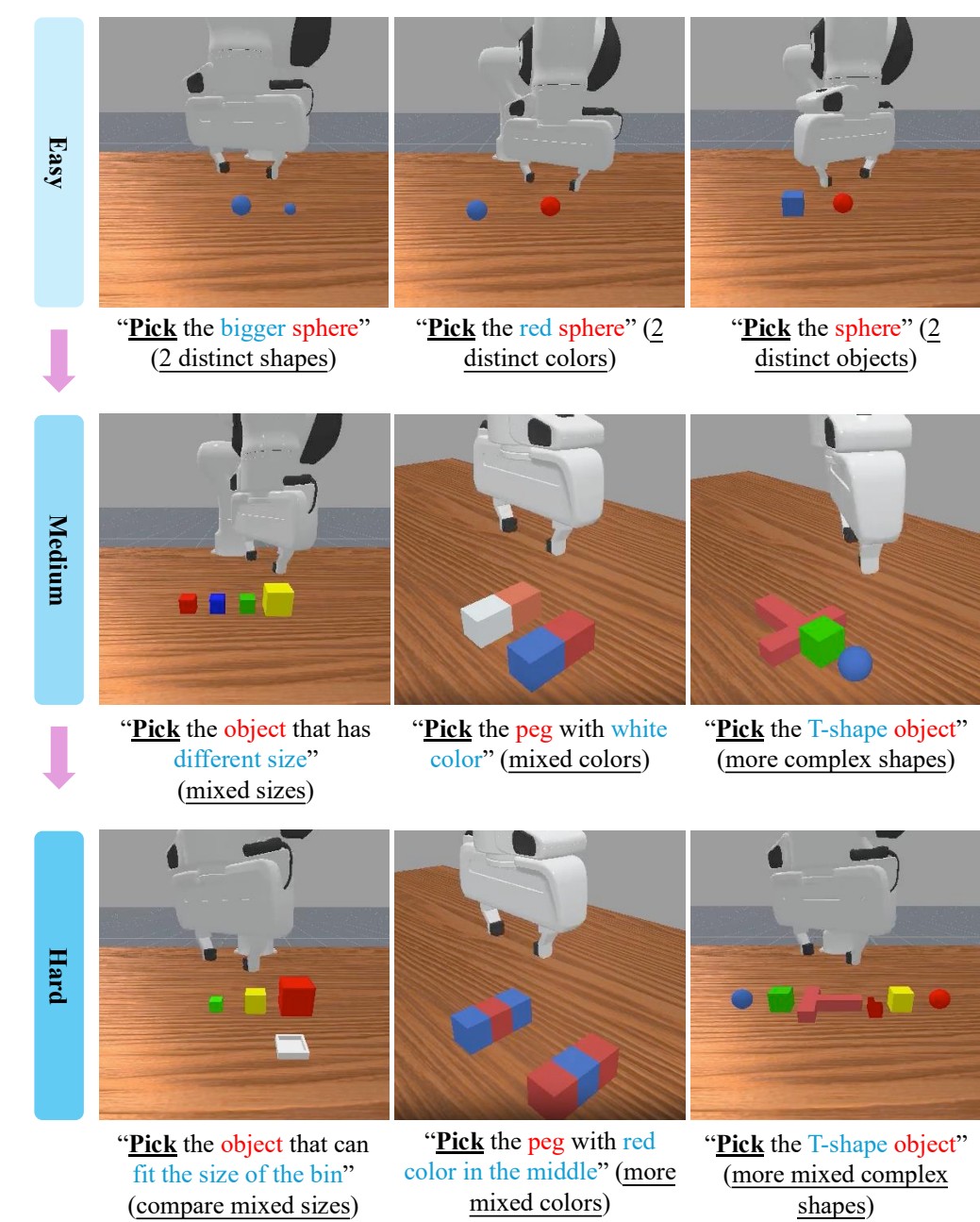

Figure 7: Visualization of NEBULA *Perception* tasks. Scenes illustrate object-level attribute discrimination with minimal control requirements. Objects are shown in green, targets in red, and contextual cues in blue. **Bold underlined** text highlights the attribute-based selection action. Tasks range from clearly distinguishable attributes to fine-grained variations that challenge visual recognition under controlled, uncluttered layouts.

based on the specified attribute, as determined by contact events with ground-truth targets. Grasp success or action quality is not evaluated, allowing task completion through mere object contact. The visualization and corresponding language commands are shown in Figure 7.

To isolate perceptual skills, all instances are designed to eliminate confounds from control or language. Objects are placed in uncluttered, reachable areas, and task goals are defined with clear visual references. Instructions use simple, unambiguous descriptions of attributes (*e.g.,* "pick up the red cube"), and environments avoid visual clutter, enabling perception to be the dominant bottleneck. A

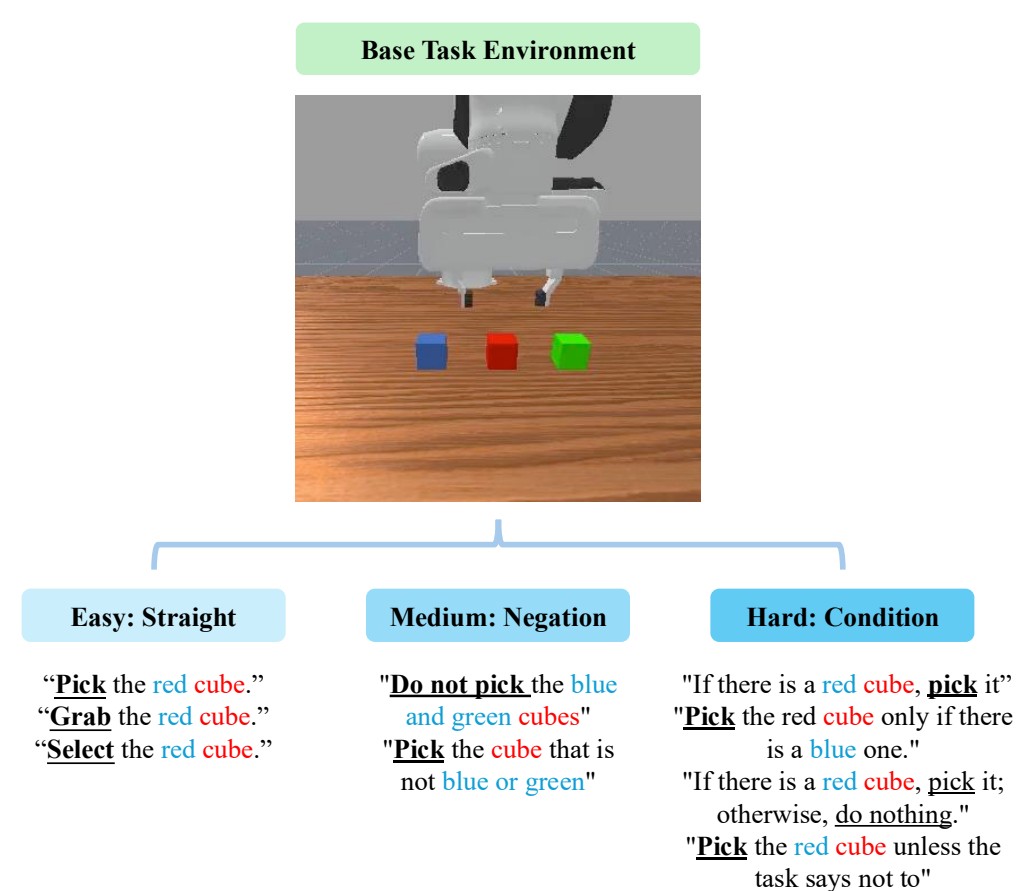

Figure 8: The *Language* in NEBULA evaluates a model's ability to interpret and act upon natural language instructions in robotic manipulation settings. red marks targets, and blue indicates contextual cues. **Bold underlined** text shows actions.

representative design choice is that agents are only required to make contact with or lift the correct object, without any requirement for trajectory precision or downstream placement. This ensures that success reflects visual identification rather than motor competence.

An example task may require the agent to identify and select the largest object among distractors of varying size. The agent succeeds by simply making physical contact with the correct object, without needing to lift, move, or manipulate it further. This removes dependence on motor execution and enables clean attribution of outcomes to perceptual capacity. Agents with poor perception may fail by selecting distractors with similar but incorrect attributes—such as picking a medium-sized object instead of the largest—or by showing indecision when faced with low contrast or partial occlusion. These behaviors reflect insufficient visual resolution or limited generalization across subtle attribute variations.

**Language**   This task family targets the agent's ability to correctly interpret and execute natural language instructions, focusing on linguistic understanding as the core competency. The key behavioral signal is whether the agent selects and manipulates the correct object or performs the correct action in accordance with the instruction's semantic intent. To ensure that language is the sole bottleneck, each scene is standardized across all task variants: objects, layouts, colors, and spatial relationships remain fixed, with only the instruction changing across samples. This removes perceptual, spatial, and control-related variance and ensures that performance differences reflect variations in language comprehension alone. The visualization and corresponding instruction examples are shown in Figure 8.

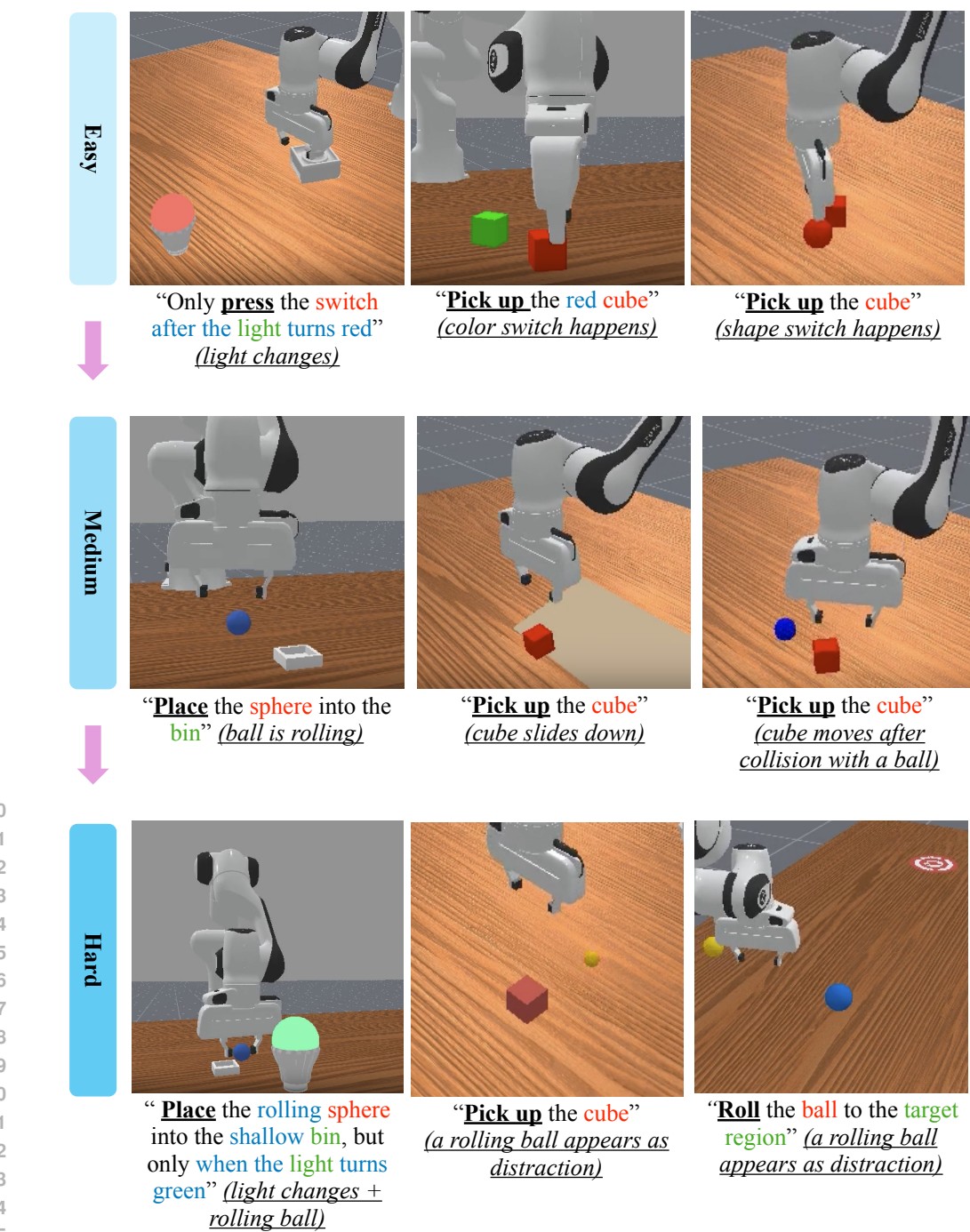

Figure 9: Visualization of NEBULA *Dynamic Adaptation* tasks. Green marks objects, red marks targets, and blue indicates contextual cues. **Bold underlined** text shows actions; *italic underlined* text gives clarifications.

The language tasks are distinguished from both Perception and Control in that the agent receives identical visual input but must succeed by correctly grounding or reasoning over language, not by recognizing attributes or executing motor skills. For example, an agent with strong perceptual capacity but poor linguistic understanding may misinterpret negation or conditionals and act on the wrong object, despite visually identifying all objects correctly.

**Dynamic Adaptation**   This capability evaluates how well an agent can operate under time-varying and non-stationary conditions. It specifically targets the agent's ability to perceive changes in the environment, track evolving states, and dynamically adjust its policy in real-time. The core skill tested is reactivity to external dynamics and internal temporal constraints. Observable behaviors include whether the agent successfully adapts its motion plans in response to shifting object positions, meets timing constraints in reactive tasks, or maintains stability under unpredictable perturbations. Successful agents exhibit behaviors such as re-planning mid-execution, anticipating changes based on motion cues, and completing tasks within the imposed temporal limits. The visualization and corresponding instructions are shown in Figure 9.

To isolate dynamic adaptation, all task scenes use visually simple setups and remove ambiguity from language or perception. Object appearances, instructions, and goals are kept constant, and the key challenge arises solely from environmental dynamics such as movement, time-triggered changes, or external forces. This ensures that performance bottlenecks can be attributed to limitations in temporal reasoning and reactive control.

Dynamic Adaptation is distinct from Control, which focuses on executing precise static action sequences, and from Perception, which emphasizes visual feature recognition. In contrast, Dynamic Adaptation requires agents to continuously monitor evolving states and revise their strategies in real time. A static perception or fixed action plan is insufficient.

A representative task involves catching a slowly rolling ball before it falls off a platform. Although the scene is visually simple and instructions are unambiguous, success requires the agent to track the ball's position, time its action appropriately, and adjust its trajectory on-the-fly. Agents with weak dynamic adaptation may exhibit delayed reactions, rigid plans that ignore motion cues, or repeated failure to anticipate predictable events.

**Spatial Reasoning**   This capability targets an agent's understanding of geometric relationships in three-dimensional space, including relative object positioning, distance, alignment, and spatial constraints across all six degrees of freedom. The core competency lies in the ability to interpret spatial descriptors and translate them into accurate physical manipulation plans. The visualization and corresponding spatial instructions are provided in Figure 10.

Observable behaviors include whether the agent can correctly infer geometric relations from language or environmental cues and execute motions that honor these constraints. Successful performance is reflected in consistent placement of objects in geometrically valid configurations, correct execution of spatial relations such as "between," "above," or "aligned with," and smooth transitions between different spatial frames of reference.

To isolate spatial reasoning from confounding factors, all tasks are built upon visually identical objects with neutral appearances and require only minimal motor precision. Instructions are phrased in simple spatial terms without attribute ambiguity. Control difficulty is standardized across tiers, and the perceptual load is minimized by removing distractors or occlusions. This ensures that the principal challenge arises from interpreting and satisfying spatial relationships, rather than perception or dexterity.

Spatial reasoning differs from Control in that it emphasizes where and how to place or manipulate an object, not merely whether the manipulation is precise. It also diverges from Perception, which deals with object recognition based on appearance, and from Language, which focuses on instruction parsing rather than spatial execution. Spatial reasoning uniquely requires constructing a mental geometric model of the scene and acting upon it.

A representative task involves stacking a medium-sized cube precisely on top of a smaller one while aligning their front faces. Although both cubes look visually identical and are easily reachable, the agent must understand both vertical and front-facing alignment to succeed. Agents lacking spatial reasoning may grasp the correct object but place it in the wrong location, misalign it rotationally, or exhibit imprecise placements that violate task constraints.

**Robustness & Generalization**   This task family assesses an agent's ability to maintain functional performance when faced with distributional shifts in object properties, scene composition, or environmental layouts that deviate from the training distribution. The core competency lies in adapting to previously unseen or perturbed conditions without requiring policy retraining or explicit fine-

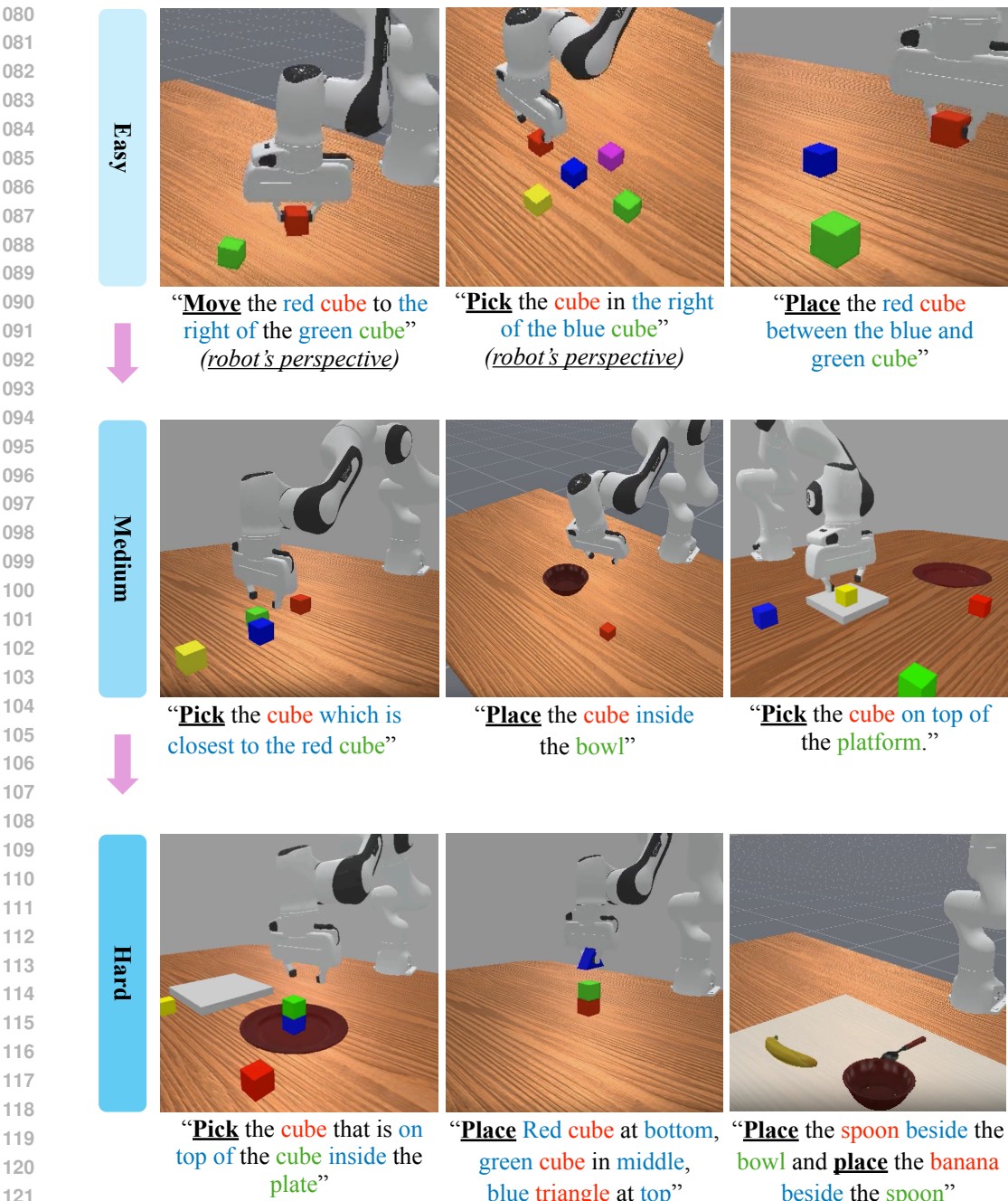

Figure 10: Visualization of NEBULA *Spatial Reasoning* tasks. Green marks objects, red marks targets, and blue indicates contextual cues. **Bold underlined** text shows actions; *italic underlined* text gives clarifications.

tuning. Difficulty scales from Easy (additional distractors in familiar scenes) to Medium (modified object attributes such as color shifts) to Hard (entirely novel scenes with out-of-distribution object categories and layouts). The visualization and corresponding language commands are demonstrated in Figure 11.

Observable signals of success include the agent's ability to ignore irrelevant distractors, generalize previously learned concepts to novel visual or structural variations, and complete tasks consistently despite altered inputs. Agents demonstrating strong robustness adapt seamlessly to variations in

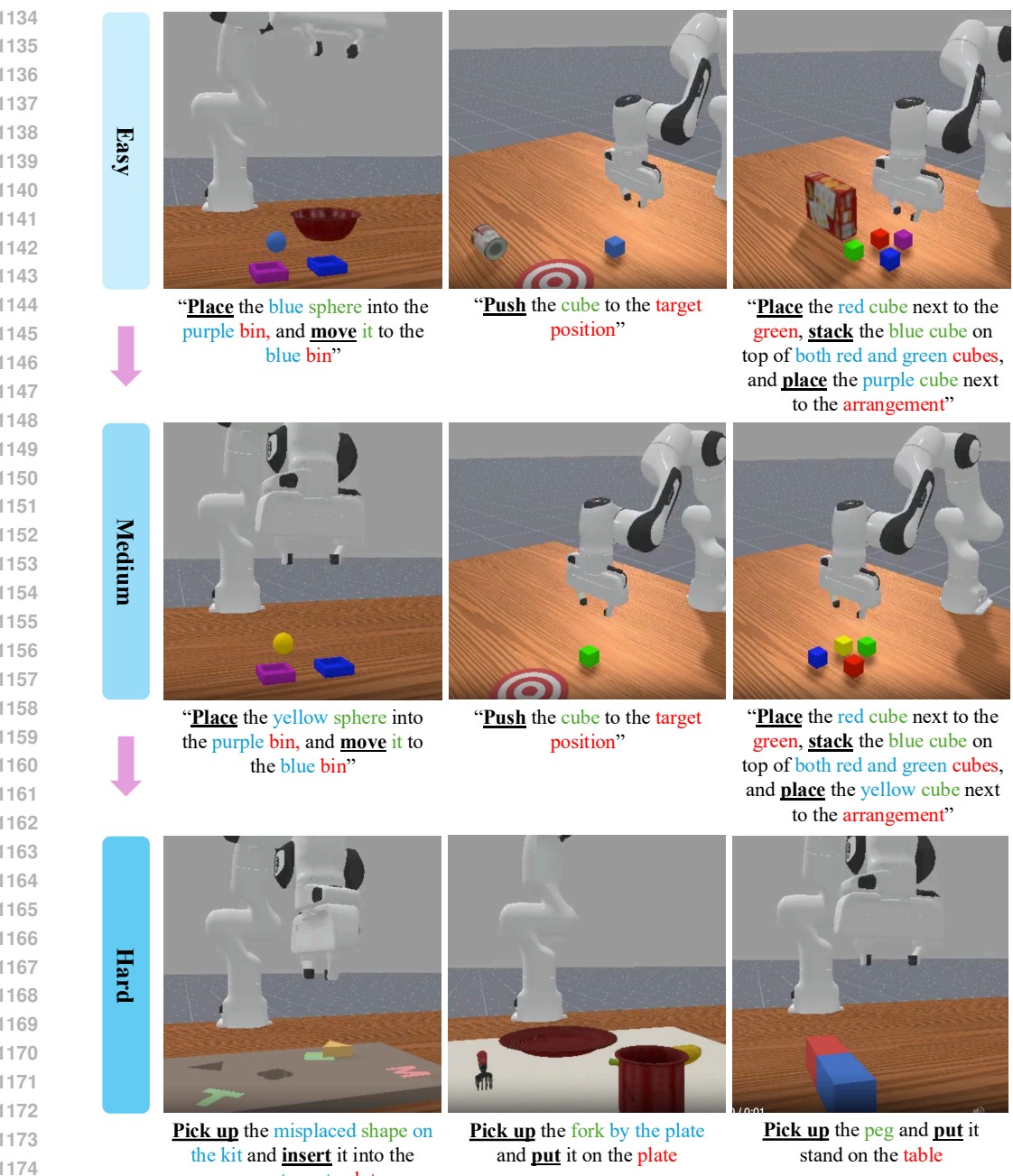

Figure 11: Visualization of NEBULA Robustness/Generalization tasks. Green marks objects, red marks targets, and blue indicates contextual cues. **Bold underlined** text shows actions.

object color, size, or scene clutter, while those with strong generalization capabilities succeed in novel environments by leveraging transferable patterns in task logic and spatial relationships.

A representative example involves a previously seen scene in which a red cube must be picked up. In the perturbed version, the cube remains the correct target but is now orange, and two new irrelevant shapes are added nearby. The instruction remains unchanged. A robust agent will attend to shape cues or structural language to identify the correct object, while a brittle agent may rely on color-matching heuristics and fail under this shift.

### A.1.3 SKILL INTERDEPENDENCE

Although embodied skills such as perception, control, language grounding, and spatial reasoning naturally interact, NEBULA's design aims to minimize these dependencies by suppressing non-target skills through controlled task construction. To empirically validate this isolation, we conduct a set of ablation-style experiments that measure how perception accuracy changes when individual non-perception factors are deliberately reintroduced. The results are summarized in Table 5.

In each experiment, the visual scene and perceptual attributes remain identical; only one additional skill requirement is introduced at a

Table 5: Success rates of GR00T-1.5 on three Perception (Easy level) tasks, comparing settings with isolated factors versus unisolated scenes with additional distractors. Results highlight the impact of scene confounding on perceptual accuracy.

| Isolated Factors | Perception | | |
| --- | --- | --- | --- |
| | PlaceBigger Sphere | Place Red Sphere | Place Sphere |
| w/ Control | 92 | 68 | 76 |
| w/ Language | 96 | 96 | 84 |
| w/ Spatial | 76 | 78 | 64 |
| w/ Dynamic | 56 | 48 | 52 |
| w/ Novel | 0 | 0 | 0 |
| ✓ | 100 | 100 | 100 |

time. In the *w/ Control* condition, agents must not only identify the target but also perform a full grasp-and-place sequence, adding precision trajectory execution. In the *w/ Language* condition, the physical interaction requirement is minimized, but the instruction becomes linguistically complex, including negation or conditional phrasing. In the *w/ Spatial* condition, objects are placed inside containers, introducing three-dimensional relational reasoning (inside/outside) while keeping the language simple. In the *w/ Dynamic* condition, objects move during execution, requiring temporal adaptation while perception itself remains trivial. Finally, the *w/ Novel* condition replaces the entire scene with unseen layouts and distractors to probe generalization stress.

The isolated setting, shown in the bottom row, removes all such confounds: the agent interacts with a familiar scene, receives only simple attribute-based language, and succeeds merely by touching the target object. Under this fully isolated configuration, GR00T-1.5B achieves 100% success across all tasks. When any single factor is reintroduced, performance drops substantially despite identical perceptual requirements. Video analysis confirms that these failures arise from issues in the added skill dimensions (*e.g.,* control instabilities, spatial misalignment, temporal mistiming, or OOD confusion), not from perception itself.

This pattern demonstrates that unrelated bottlenecks can obscure performance in entangled settings, and it empirically supports NEBULA's controlled-variable design. By suppressing non-target competencies, each capability task offers a cleaner, more interpretable probe into the skill it is meant to measure.

However, we also acknowledge that in certain tasks, complete separation of skill components is inherently difficult. For example, spatial reasoning often co-occurs with manipulation constraints. Similarly, tasks involving dynamic object trajectories inevitably entangle temporal anticipation with control feedback. While we mitigate such overlap by reducing non-target skill demands (*e.g.,* using fixed grasp positions to minimize control complexity in spatial tasks), some degree of interdependence remains unavoidable. Rather than eliminating such cases, our design explicitly captures this spectrum of coupling, enabling systematic evaluation across both isolated and entangled conditions. This not only ensures realism but also offers insight into how skill bottlenecks emerge under different task compositions.

### A.1.4 CAPABILITY TEST RESULT

Figure 12 presents the quantitative results for the capability tests. A detailed analysis of these findings is provided in Section 4.2.

### A.2 STRESS TEST TASKS

This section details the specifications for NEBULA's Stress Test suite, designed to isolate and quantify system robustness under targeted operational constraints. Unlike aggregate robustness metrics, each test functions as a single-indicator probe, systematically varying a specific stressor (*e.g.,* latency, sensor noise, or visual distortion) across three calibrated intensity levels ($v_1$–$v_3$). This tiered

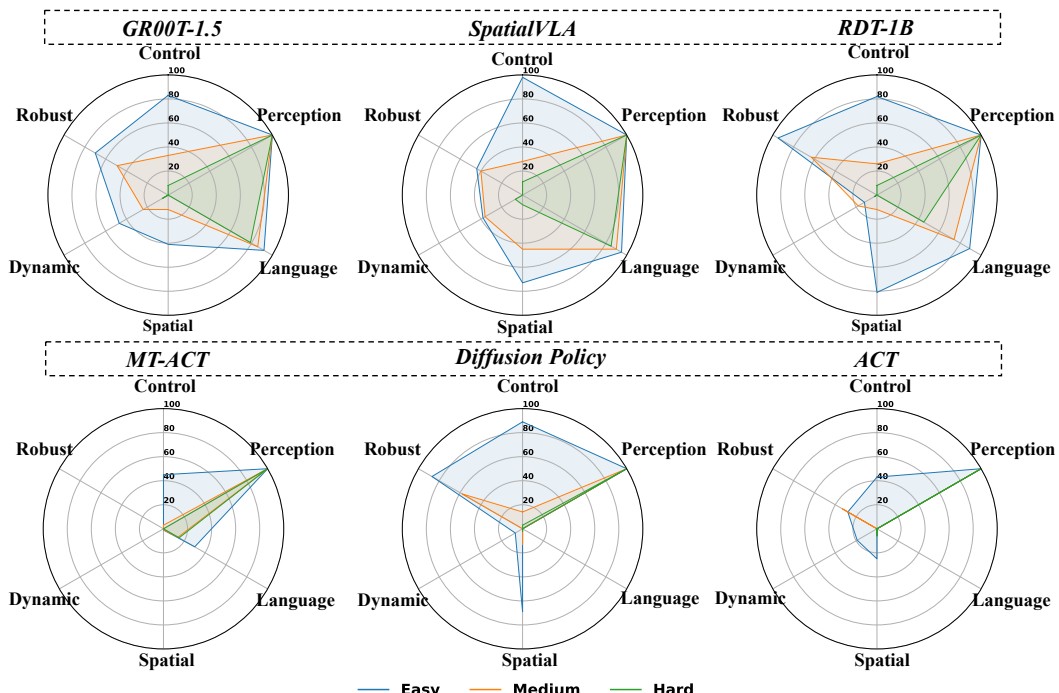

Figure 12: Performance of six agents across NEBULA's six capability axes: Control, Perception, Language, Dynamic Adaptation, Spatial Reasoning, and Robustness. Each chart displays success rates (%) for a specific model (GR00T-1.5, SpatialVLA, RDT-1B, MT-ACT, Diffusion Policy, ACT) across three difficulty tiers: Easy (blue), Medium (orange), and Hard (green). The radial axes represent the capability dimensions, with the distance from the center indicating the success rate.

structure enables precise stress-response profiling, allowing researchers to identify explicit performance boundaries and separate inherent capability limitations from deployment-induced fragility.

### A.2.1 SENSOR-LEVEL STRESS TEST

This subsection details the Perceptual Degradation Stressors, a suite of six targeted perturbations designed to evaluate agent robustness against sensor anomalies common in physical hardware. It is crucial to distinguish these stress tests from the Perception capability family defined in Appendix A.1. While the Capability axis evaluates the agent's semantic understanding of the environment (*e.g.,* distinguishing object attributes or resolving clutter), this suite targets the fidelity of the sensory signal itself. By simulating distinct hardware failure modes, ranging from stochastic signal noise to geometric distortion, we isolate low-level visual processing bottlenecks and quantify the degradation of policy performance under non-ideal sensing conditions.

**Poisson-Gaussian Noise** To simulate realistic sensor degradation, we inject heteroscedastic Poisson-Gaussian noise into the RGB observations. This mixed model captures two fundamental components of digital imaging noise: signal-dependent **photon shot noise** and signal-independent **electronic read noise**. For a normalized pixel intensity $\hat{I} = I/255$ and a severity parameter $\Lambda$ (denoted as `lam` in our codebase), we sample a Poisson component $P \sim \text{Poisson}(\hat{I} \cdot \Lambda)$ to model quantum fluctuations, and add independent Gaussian noise $G \sim \mathcal{N}(0, \Lambda^2)$ to model thermal fluctuations. This formulation effectively mimics artifacts characteristic of low-light environments or high-gain ISO settings, allowing us to evaluate policy robustness against complex, realistic visual corruption. An example is shown in Figure 13.

**Rolling-shutter Distortion** To simulate the geometric skew characteristic of CMOS sensors with sequential line readout, we apply a row-dependent horizontal shear to the RGB observations. This distortion models the temporal misalignment that occurs when the sensor's readout speed is compa-

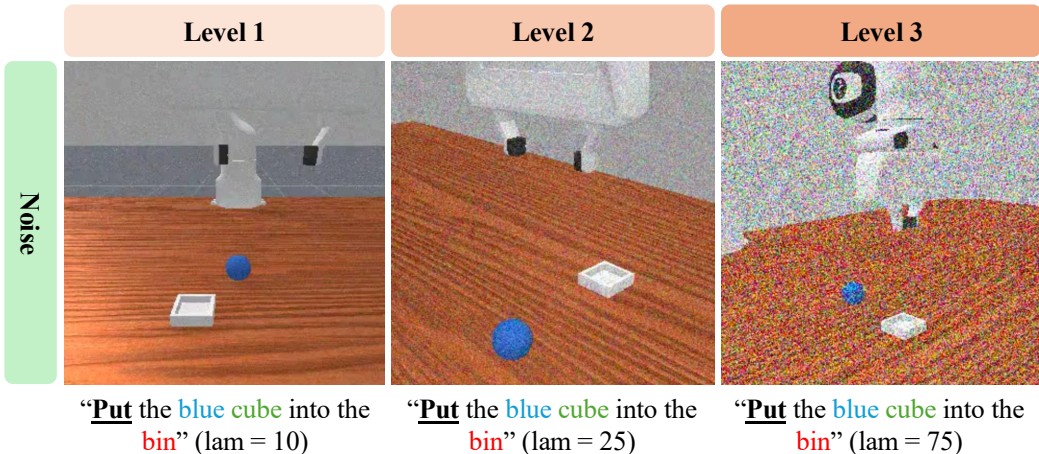

Figure 13: **Poisson-Gaussian Noise**. Visualization of a NEBULA Poisson–Gaussian noise–induced task. The severity of corruption is controlled by the parameter `lam`. Green marks objects, red marks targets, and blue indicates contextual cues.

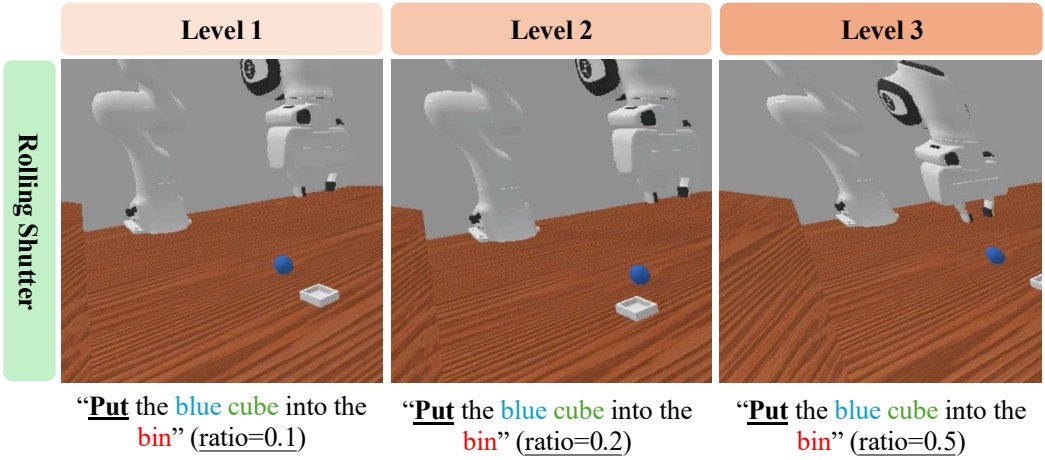

Figure 14: **Rolling-shutter Distortion**. Visualization of a NEBULA Rolling-shutter Distortion–induced task. The severity of distortion are controlled by the parameter `ratio`. Green marks objects, red marks targets, and blue indicates contextual cues.

rable to object motion. Formally, for an image of width $W$ and height $H$, the horizontal offset $\Delta x$ for a given row $y$ is modeled as $\Delta x(y) = W \cdot \gamma \cdot \mathcal{F}(y/H)$, where $\gamma \in [0, 1]$ represents the maximum distortion fraction (denoted as `ratio`) and $\mathcal{F}(\cdot)$ is a shaping function (defaulting to a square-root profile to model non-linear readout accumulation). This formulation effectively reproduces the "jello effect" and geometric warping inherent in dynamic scenes captured by rolling-shutter hardware, enabling the evaluation of VLA agents under temporally inconsistent visual inputs. An example is shown in Figure 14.

**Light Flicker induced**  To evaluate robustness against high-frequency radiometric instability, we simulate the stroboscopic interference effects caused by the interaction between alternating current (AC) illumination sources and the sensor's rolling shutter readout. We model this periodic exposure variation by applying a row-dependent sinusoidal modulation to the RGB input. Formally, for an image of height $H$, the modulation factor for a given row $y$ is defined as $\mathcal{M}(y) = 1 + \alpha \cdot \sin(2\pi \cdot \phi \cdot \frac{y}{H})$, where $\alpha$ represents the modulation amplitude (fixed at $0.1$) and $\phi$ denotes the flicker frequency parameter. The observed image is computed via element-wise multiplication $I_{obs} = \text{clamp}(I_{orig} \odot \mathcal{M}, 0, 255)$. This perturbation introduces dynamic, spatial-frequency dependent exposure artifacts,

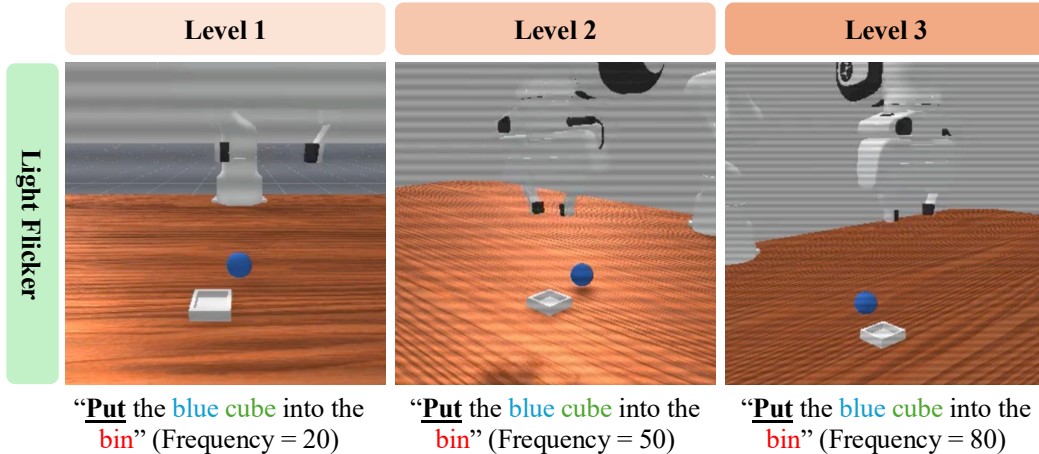

| Level 1 | Level 2 | Level 3 |
| --- | --- | --- |
| "**Put** the blue cube into the bin" (Frequency = 20) | "**Put** the blue cube into the bin" (Frequency = 50) | "**Put** the blue cube into the bin" (Frequency = 80) |

Figure 15: **Light Flicker**. Example of a NEBULA task with row-wise sinusoidal illumination modulation controlled by flicker frequency. Green marks objects, red marks targets, and blue indicates contextual cues.

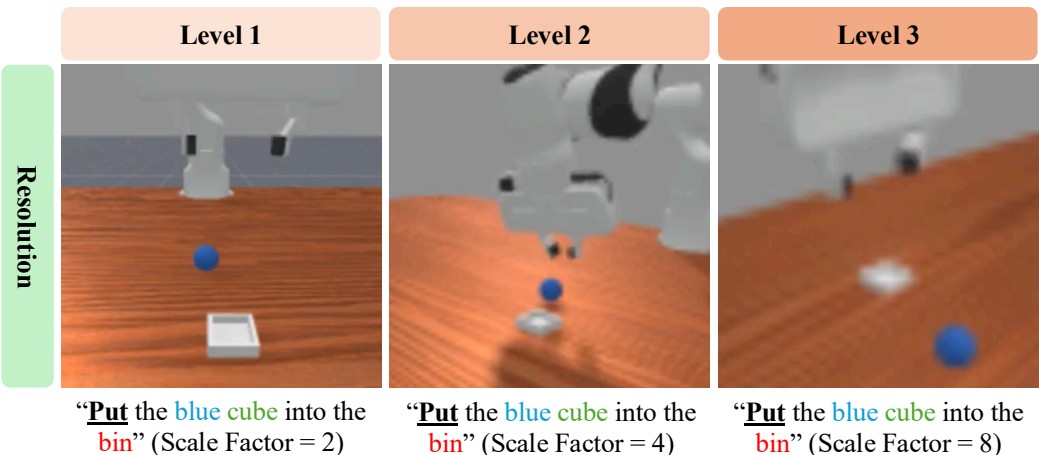

| Level 1 | Level 2 | Level 3 |
| --- | --- | --- |
| "**Put** the blue cube into the bin" (Scale Factor = 2) | "**Put** the blue cube into the bin" (Scale Factor = 4) | "**Put** the blue cube into the bin" (Scale Factor = 8) |

Figure 16: **Resolution degradation**. Example of a NEBULA task with reduced effective resolution, controlled by the `scale factor`. Green marks objects, red marks targets, and blue indicates contextual cues.

challenging the agent to maintain state estimation stability under non-stationary lighting conditions. An example is shown in Figure 15.

**Resolution Degradation**   To assess policy robustness against spatial information loss, we simulate the effects of signal bandwidth constraints and low-fidelity sensor quantization. We model this via a downsample-upsample pipeline that effectively reduces the Nyquist frequency of the visual input. Formally, an input image $I \in \mathbb{R}^{H \times W \times C}$ is first subjected to area-based pooling with a kernel size $k$ (corresponding to the scale factor), reducing the spatial dimensionality to $H/k \times W/k$. This latent low-resolution representation is then reconstructed to the original dimensions via bilinear interpolation. This process acts as a non-ideal low-pass filter, suppressing high-frequency spatial features to mimic the artifacts of heavy transmission compression or inferior optical resolving power. An example is shown in Figure 16.

**Frame Drops**   To assess policy resilience against temporal observation sparsity and signal interruption, we simulate the effects of stochastic data loss characteristic of unstable transmission channels. We model this process by subjecting the RGB stream to a Bernoulli dropout mechanism.

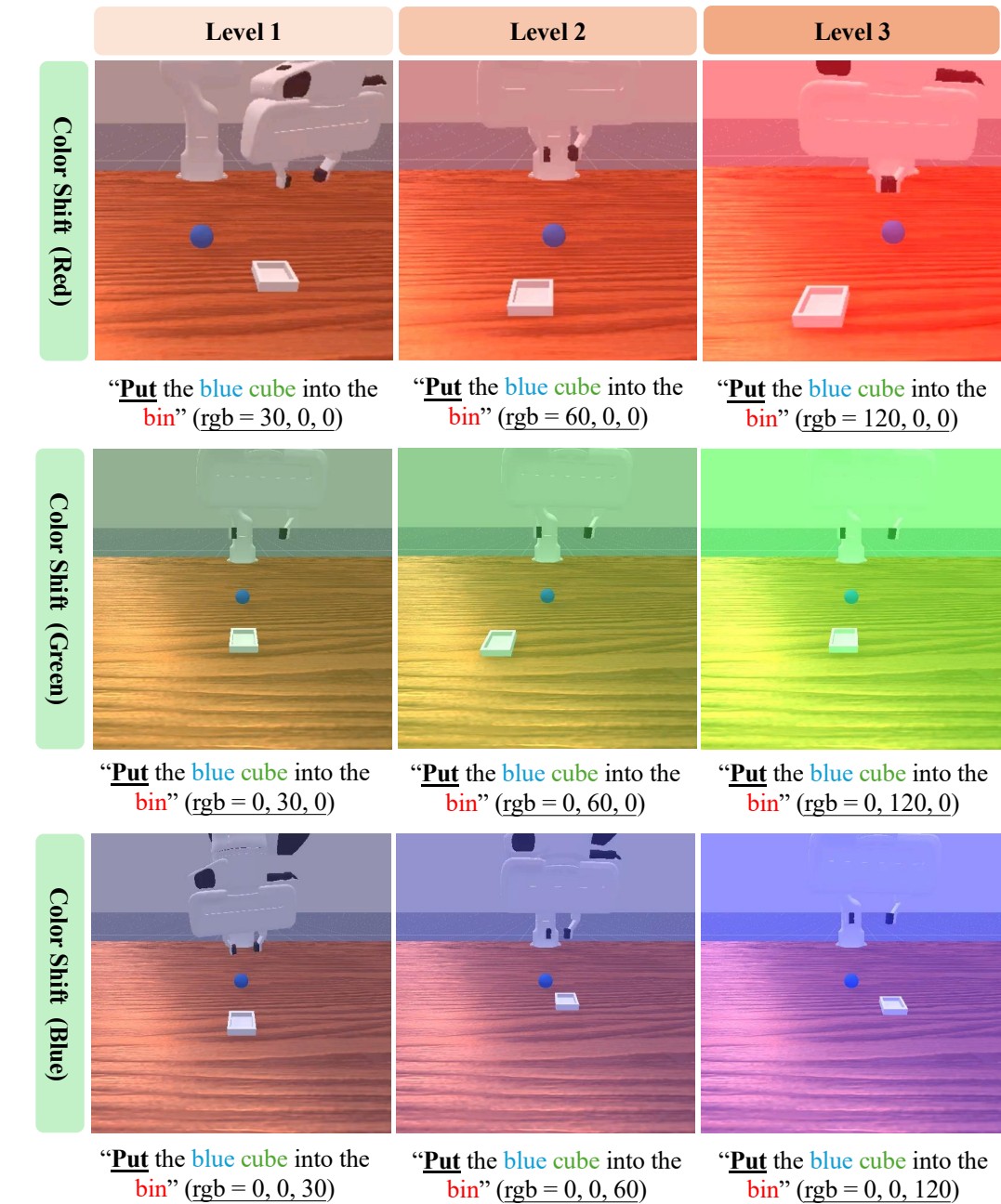

Figure 17: **Color cast**. Example of a NEBULA task with a per-channel RGB bias applied uniformly across the image. The red, green, and blue channels are controlled through a three-dimensional RGB bias, producing distinct color-shift patterns. Green marks objects, red marks targets, and blue indicates contextual cues.

Formally, at each timestep $t$, the observation $I_t$ is preserved with probability $1 - \rho$ or replaced by a null (zero-value) tensor with probability $\rho$ (corresponding to the drop_rate). This perturbation mimics critical deployment failures such as network packet loss, buffer overflows, or hardware-level sensor blackouts, challenging the agent to maintain state estimation consistency despite incomplete temporal support.

**Color Cast** To evaluate policy invariance to global radiometric distortions, we simulate the chromatic shifts caused by automatic white balance (AWB) failures or sensor spectral sensitivity drift.

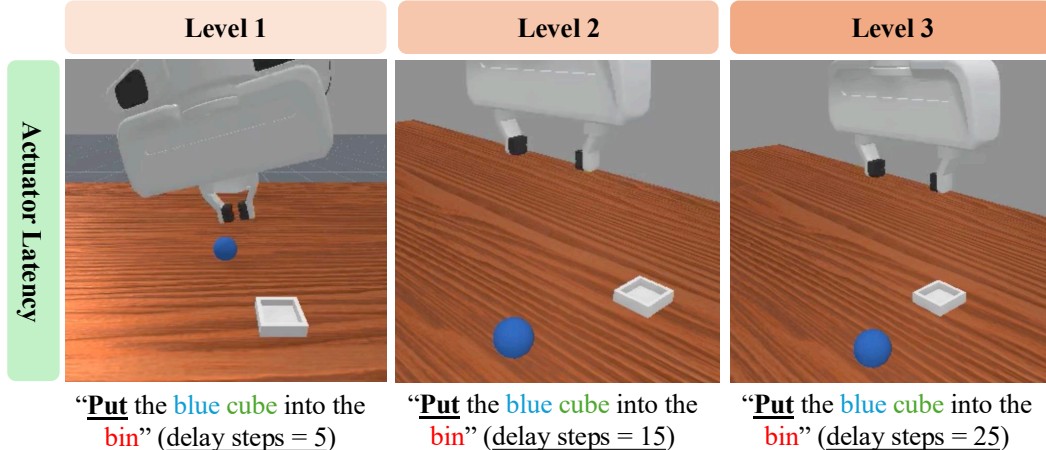

**Actuator Latency**

| Level 1 | Level 2 | Level 3 |

"**Put** the blue cube into the bin" (delay steps = 5)     "**Put** the blue cube into the bin" (delay steps = 15)     "**Put** the blue cube into the bin" (delay steps = 25)

Figure 18: **Actuator Latency**. Example illustrating delayed actuation for the same frame and timestamp, controlled by the delay_steps parameter.

We model this as a spatially uniform, channel-wise additive bias. Formally, given an input pixel vector $\mathbf{x} \in \mathbb{R}^3$ (RGB), we apply a bias vector $\mathbf{b} \in \mathbb{R}^3$ (e.g., $[30, 0, 0]^\top$) to generate the corrupted observation $\mathbf{x}' = \text{clamp}(\mathbf{x} + \mathbf{b}, 0, 255)$. This perturbation introduces systematic hue and saturation offsets, challenging the agent to maintain robust feature recognition despite distributional shifts in the color space characteristic of uncalibrated or aging optical hardware. An example is shown in Figure 17.

A.2.2 EXECUTION-LEVEL STRESS TEST

This subsection details the Execution-Level Stressors, a suite of protocols designed to evaluate the integrity of the downstream control and communication pipeline. While perceptual tests target input fidelity, these tests assess the robustness of the policy's actuation output. By quantifying trajectory smoothness (Stability) and simulating transmission anomalies (Actuator Latency, Command Packet Loss), we measure the agent's ability to maintain precise physical control despite the temporal lags, signal interruptions, and motor noise characteristic of real-world hardware interfaces .

**Stability Score** Stability scores quantifies trajectory smoothness by measuring action variation between consecutive timesteps. Given an action sequence $\{a_0, a_1, ..., a_t\}$, the score is computed as:

$$\text{Stability} = \exp\left(-\frac{1}{T-1}\sum_{t=1}^{T} ||\mathbf{a}_t - \mathbf{a}_{t-1}||_2\right)$$

where $||\mathbf{a}_t - \mathbf{a}_{t-1}||_2$ represents the $L2$ norm of action changes between neighboring timesteps and the exponential decay of mean action changes yields a normalized score $\in [0, 1]$, with 1 indicating perfect stability.

The test evaluates three precision levels using Control family stack cube tasks (Easy to Hard), where success inherently requires smooth trajectories for stable object manipulation as $v1$ to $v3$ with increasing complexity. This metric reveals whether VLA policies generate stable control signals suitable for physical deployment, distinguishing smooth execution from erratic behaviors that could damage hardware or cause task failure. Due to exponential scaling, small numerical differences reflect significant changes in underlying action deviation.

**Actuator Latency** To evaluate policy stability under non-instantaneous actuation, we introduce a fixed temporal delay (dead time) between command issuance and execution. Formally, the executed action at time $t$ is given by $a_{exec}^{(t)} = a_{issued}^{(t-k)}$, where $k$ represents the delay horizon maintained via a FIFO buffer. We evaluate this stressor across two distinct regimes: (1) **Static Tasks**, where

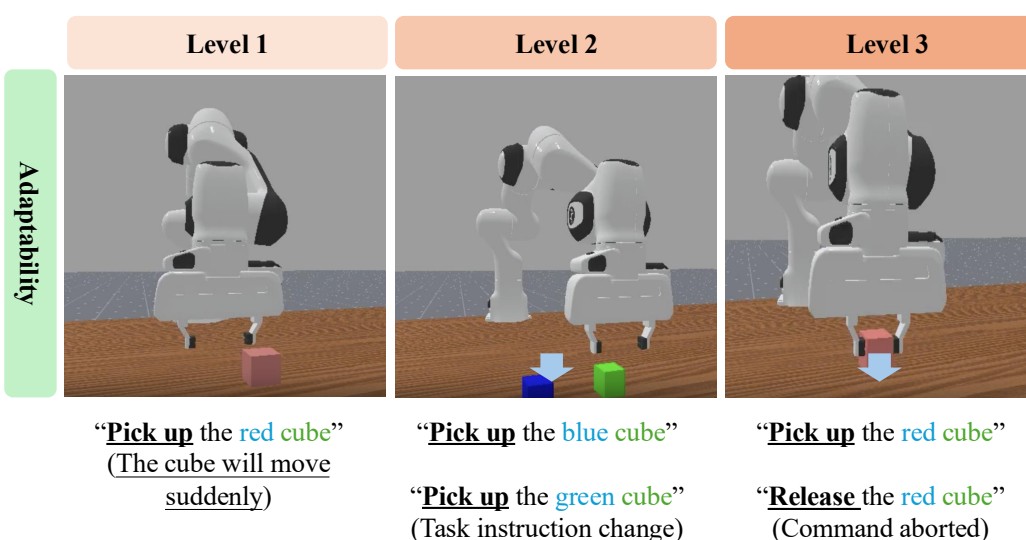

Figure 19: Visualization of NEBULA Adaptability tasks. Green marks objects, red marks targets, and blue indicates contextual cues. **Bold underlined** text shows actions; *italic underlined* text gives clarifications.

objects are stationary, probing the agent's susceptibility to feedback-induced oscillations and overshooting due to delayed error correction; and (2) **Dynamic Tasks**, where objects move, challenging the agent's predictive control capabilities and bandwidth when tracking targets under significant transport delays. This setup mirrors critical deployment constraints such as network latency or electromechanical response times. An example is shown in Figure 18.

**Command Packet Loss**   To assess policy robustness against intermittent transmission failures, we simulate a lossy control channel characterized by stochastic packet erasure. We model this using a Bernoulli process coupled with a **Zero-Order Hold (ZOH)** reconstruction strategy. Formally, the executed action $a_{exec}^{(t)}$ is defined recursively: with probability $1 - \rho$, the current command is applied ($a_{exec}^{(t)} = a_{issued}^{(t)}$); with probability $\rho$ (corresponding to the drop_rate), the previous action is repeated ($a_{exec}^{(t)} = a_{exec}^{(t-1)}$). This formulation mimics the "hold-last-sample" behavior common in networked control systems facing UDP packet loss or wireless interference, challenging the agent's ability to maintain trajectory smoothness despite discrete signal discontinuities.

### A.2.3    Cognitive-Level Stress Test

This subsection details the Cognitive-Level Stressors, a suite of protocols designed to evaluate the computational efficiency and reasoning plasticity of the policy's decision-making core. Distinct from the signal degradation tested in the Sensor and Execution layers, these tests probe the intrinsic architectural limitations of the agent itself. By quantifying Inference Frequency and assessing Adaptability under non-stationary objectives, we determine whether the agent possesses the requisite processing bandwidth and logical flexibility to sustain closed-loop control in dynamic, real-time environments.

**Inference Frequency**   Inference frequency measures the rate at which an agent generates control actions in hertz. This metric directly impacts an agent's ability to respond to dynamic environments and maintain smooth control. The test evaluates inference frequency under three scenarios: $v_1$ tests slow and uniform movements; $v_2$ tests alternating movement at medium speed; $v_3$ tests fast irregular movements. Performance degradation across tiers reveals how VLA models handle increasing computational demands, exposing whether failures stem from insufficient inference speed and model architecture limitations.

**Adaptability** Adaptability measures an agent's ability to adjust its behavior in response to environmental changes, task interruptions, or modified objectives during execution. The test evaluates the model's performance across three scenarios: $v_1$ tests response to object displacement where the target suddenly moves to a new position; $v_2$ introduces mid-task instruction changes, requiring the agent to switch between objectives (*e.g.,* "Pick up the blue cube" → "Pick up the green cube"); $v_3$ demands rapid re-planning under sequential instructions (e.g., "Pick up the cube" → "Release the cube"). This progression assesses whether VLA policies can maintain task coherence under dynamic conditions, distinguishing reactive agents that gracefully handle perturbations from rigid controllers that fail when initial assumptions are violated. The visualization and corresponding language commands are demonstrated in Figure 19.

A.2.4    STRESS TEST RESULT

This section presents the comprehensive quantitative analysis of the NEBULA Stress Test suite. We report the performance of evaluated baselines across the three calibrated severity tiers ($v_1-v_3$) for all defined cognitive, sensor, and execution stressors. By quantifying the rate of performance degradation under increasing perturbation, these results empirically map the operational boundaries of each agent. This sensitivity analysis reveals distinct robustness profiles and highlights critical reliability bottlenecks, such as latency-induced failure or sensor fragility, that remain hidden under nominal evaluation conditions.

**Sensor-Level Stress Test** The evaluation of Sensor Degradation Stressors (Fig 20) reveals a stark stratification in visual robustness across architectures, exposing critical vulnerabilities in how current VLA agents process degraded sensory signals. GR00T-1.5 consistently demonstrates superior resilience compared to baselines, maintaining non-trivial success rates even under high-severity ($v_3$) perturbations, whereas Diffusion Policy (DP) exhibits a near-total collapse across all sensor defects. This fundamental performance gap suggests that the visual representations in standard behavior-cloning policies like DP lack the necessary invariance to handle even minor distributional shifts in signal quality, rendering them brittle in deployment scenarios involving camera noise or transmission loss. In contrast, the foundational pre-training of GR00T appears to confer a significant robustness advantage, particularly in tasks involving global information loss. For instance, under Resolution Degradation and Light Flicker, GR00T retains high performance, dropping only from 92% to 66% and 88% to 66% respectively at the most severe level. This indicates that its visual encoder possesses effective scale invariance and temporal consistency, allowing it to operate reliably despite reduced signal bandwidth or radiometric instability.

However, specific failure modes highlight universal bottlenecks in the current generation of vision-based controllers. While agents show relative resilience to global downsampling (Resolution), they are highly sensitive to high-frequency pixel-level corruption and geometric distortion. Under Poisson & Gaussian Noise, even the top-performing GR00T suffers a steep performance decline from 68% ($v_1$) to 20% ($v_3$), while SpatialVLA and RDT-1B crash to success rates below 10%. Rolling Shutter distortion proves to be the most devastating stressor; nearly all models, including GR00T, exhibit a "failure cliff" where performance evaporates at higher distortion ratios ($v_2, v_3$). This suggests that while current Vision Transformers (ViTs) and encoders are robust to semantic abstractions, they rely heavily on precise spatial geometry and clean high-frequency texture information. The geometric shear introduced by rolling shutter artifacts disrupts the spatial tokenization process essential for manipulation, indicating that future architectures must explicitly incorporate augmentation strategies or inductive biases to account for the physical artifacts of CMOS sensor readout and low-light acquisition.

**Execution-Level Stress Test** The quantitative analysis (Fig 21) of execution stressors exposes a fundamental trade-off between trajectory smoothness and dynamic robustness, while revealing a universal fragility to network anomalies. As illustrated in the Stability Score results, Diffusion Policy (DP) dominates in trajectory quality, maintaining scores above 0.97 even under maximum stress ($v_3$); this confirms that diffusion-based action generation inherently produces superior, jerk-free control signals compared to VLA-based autoregression. In contrast, SpatialVLA exhibits significant degradation at high stress levels (dropping to 0.86), indicating that monolithic VLA architectures struggle to maintain temporal coherence in their action outputs when pushed to their computational limits. However, this stability advantage proves insufficient in non-ideal transmission conditions.

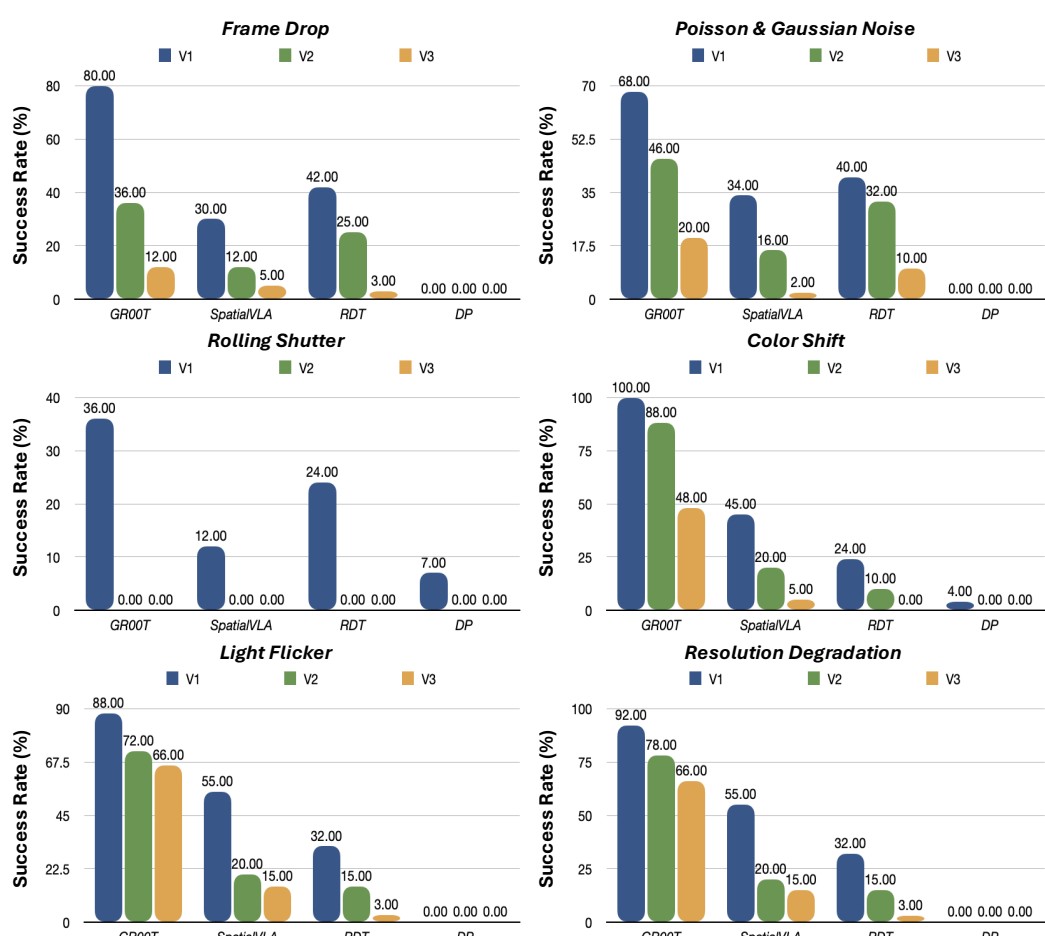

Figure 20: **Sensor-Level Stress Test Results**. Quantitative evaluation of four agents (GR00T, Spatial­VLA, RDT, DP) across six perceptual degradation stressors: Frame Drop, Poisson & Gaussian Noise, Rolling Shutter, Color Shift, Light Flicker, and Resolution Degradation. The vertical axis represents the Success Rate (%), while the colored bars denote the three stress severity levels ($v_1$, $v_2$, $v_3$), representing increasing perturbation intensity. Higher score represent better performance.

While all agents maintain high success rates ($> 90\%$) under Static Actuator Latency, where the stationary nature of the task allows the policy to eventually converge despite delays, performance collapses catastrophically under Dynamic Actuator Latency. The universal failure (0% success) across all models in the dynamic regime indicates that current agents operate as purely reactive controllers; they lack the predictive horizon necessary to compensate for dead time when tracking non-stationary targets. Furthermore, the Command Packet Loss test demonstrates extreme system fragility, with a 0% success rate across all architectures. This implies that state-of-the-art policies rely heavily on uninterrupted, perfect control loops and lack the internal robustness to recover from the stochastic signal dropouts characteristic of real-world wireless deployment.

**Cognitive-Level Stress Test**  The evaluation of Cognitive-Level Stressors uncovers a profound disparity in computational efficiency that directly dictates agent reactivity. As evidenced by the Inference Frequency results, GR00T-1.5 establishes a distinct performance tier, maintaining a high control frequency of approximately 17 Hz even under complex motion constraints ($v_3$). In stark contrast, monolithic transformer architectures like SpatialVLA and RDT-1B, as well as the iterative Diffusion Policy, suffer from severe throughput limitations, operating at roughly 2 Hz, 5 Hz, and 1.5 Hz respectively. This order-of-magnitude gap indicates that while large-scale autoregressive baselines may excel at static planning, their inference latency creates a prohibitive bottleneck

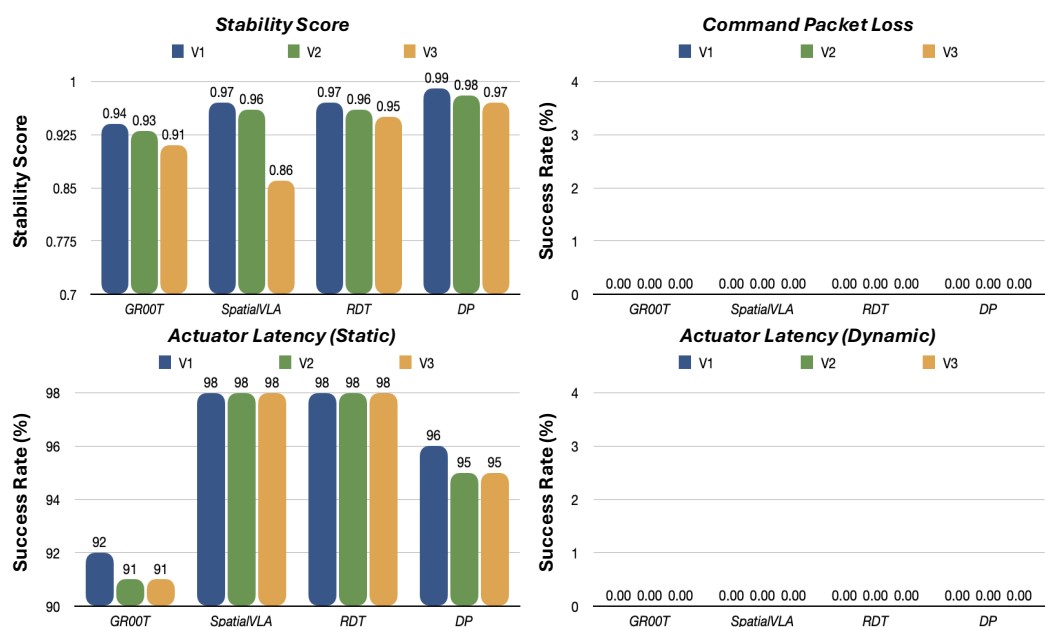

Figure 21: **Execution-Level Stress Test Results**. Evaluation of four agents (GR00T, SpatialVLA, RDT, DP) across three actuation integrity stressors: Stability Score, Command Packet Loss, and Actuator Latency (evaluated in both Static and Dynamic regimes). The bars represent performance metrics across three calibrated stress levels ($v_1, v_2, v_3$). Higher score represent better performance.

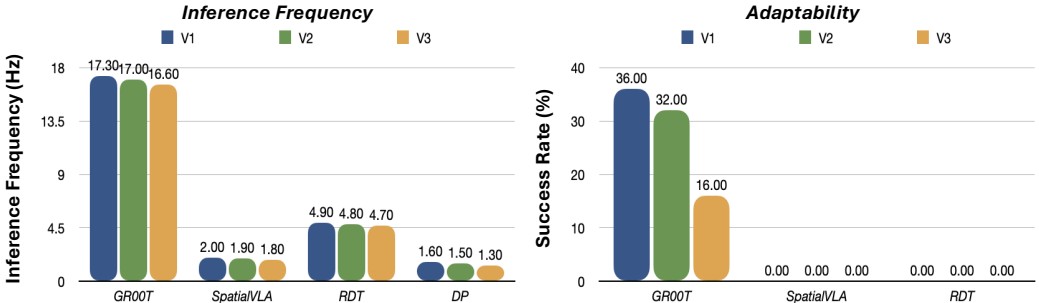

Figure 22: **Cognitive-Level Stress Test Results**. Quantitative evaluation of agent computational efficiency and reasoning plasticity. The left chart reports Inference Frequency (Hz) across three motion complexity levels ($v_1, v_2, v_3$), while the right chart reports Adaptability success rates (%) under dynamic goal shifts. Note that Diffusion Policy (DP) is omitted from the Adaptability plot due to its lack of language conditioning capabilities.

for real-time interaction, effectively decoupling the agent's decision cycle from the environment's physical dynamics.

This frequency deficit manifests as a binary failure mode in the Adaptability test. GR00T is the sole architecture to demonstrate meaningful reasoning plasticity, achieving a 36% success rate at $v_1$, although this degrades to 16% under maximal stress ($v_3$), highlighting the inherent difficulty of rapid context switching even for fast agents. Conversely, SpatialVLA and RDT-1B exhibit a complete inability to adapt, flatlining at 0% success across all severity levels. This provides empirical validation for the "Real-Time Barrier" hypothesis: despite possessing strong semantic planners (as evidenced in Capability tests), these agents are physically unable to close the perception-action loop fast enough to intercept moving targets or re-ground updated instructions. Their internal reasoning latency exceeds the timescale of environmental perturbations, rendering them functionally blind to dynamic goal shifts.

Table 6: Comparison of NEBULA and existing single-arm manipulation benchmarks across task design and evaluation protocols. NEBULA uniquely supports both capability evaluation and stress testing. Unlike prior benchmarks, it adopts a dual-axis protocol that evaluates skills and stress responses separately, ensuring each score reflects a specific factor. Other benchmarks mostly report task-level success rate without isolating capabilities or stress conditions, limiting diagnostic insight.

| Benchmark | Task Design | | | | Data Design | |
|---|---|---|---|---|---|---|
| | Task Families | Language | Tiered Difficulty | Evaluation | # Modality | # View |
| ManiSkill | Multiple | ✗ | ✗ | TSR | 1 | 3 |
| RLBench | Multiple | ✗ | ✗ | TSR | 1 | 2 |
| FurnitureBench | Furniture | ✗ | ✗ | TSR | 1 | 2 |
| BridgeDataV2 | Pick/Place | ✗ | ✗ | TSR | 1 | 2 |
| Meta-World | Multiple | ✗ | ✗ | TSR | 1 | 2 |
| FrankaKitchen | Kitchen-related | ✗ | ✗ | TSR | 1 | 2 |
| CLVIN | Visual Reasoning | ✓ | ✗ | TSR | 1 | 2 |
| ALFRED | Compositional | ✓ | ✗ | TSR | 1 | 2 |
| LIBERO | Language | ✓ | ✓ | TSR | 1 | 2 |
| VLABench | Realistic | ✓ | ✓ | TSR | 1 | 2 |
| **NEBULA (Ours)** | 6 Capabilities | ✓ | ✓ | DAE | 3 | 6 |

**Notes**: *SR* represents *Success Rate*, *TSR* represents *Task-level Success Rate*, *DAE* represents *Dual-Axis Evaluation*

## A.3 BENCHMARK COMPARISON

Table 6 indicates that NEBULA uniquely implements dual-axis evaluation (DAE) that separates capability assessment from stress testing, while all other benchmarks report only task-level success rates. Table 6 also highlights NEBULA's comprehensive data collection: three modalities (RGB, depth, segmentation) and six camera viewpoints versus the single modality and 1-2 views standard in other benchmarks. Only LIBERO and VLABench match NEBULA's tiered difficulty structure, though neither provides the diagnostic isolation of specific capabilities that NEBULA's six distinct task families enable.

## A.4 EXPERIMENTAL SETTING

All capability tasks in the Alpha dataset except are used for fine-tuning. To ensure fair comparison across baselines, we adapted only the data loading code to make NEBULA-Alpha compatible with each model's training pipeline, while strictly following the official fine-tuning protocols released by the authors on GitHub. All hyperparameters, loss functions, and architectural configurations remain unchanged from their original implementations.

**Evaluation Protocol:** We evaluate 6 capability families (Control, Perception, Language, Spatial, Dynamic, Robustness), each containing 3 difficulty levels (Easy, Medium, Hard), and 11 stress test families, each with 3 pressure levels (v1, v2, v3). Each task variant is tested over 25 episodes with a maximum episode length of 400 steps. Episodes terminate early upon task success; otherwise, they run until the 400-step limit is reached.

**Computational Resources:** Fine-tuning resources for all baselines are summarized in Table 7.

**Stress Test Parameters:** Complete parameter specifications for all stress tests across three pressure levels are detailed in Table 8.

Table 7: Fine-tuning computational resources for evaluated baselines.

| Model | GPU Configuration | Training Time |
|---|---|---|
| GR00T-1.5 | 1× A6000 | 58 hours |
| RDT-1B | 4× A100 | 202 hours |
| Diffusion Policy | 1× A6000 | 76 hours |
| SpatialVLA | 1× A6000 | 98 hours |
| MT-ACT | 1× A6000 | 20 hours |
| ACT | 1× A6000 | 16 hours |

Table 8: Stress test parameter specifications across three pressure levels (v1, v2, v3).

| Stress Test | v1 | v2 | v3 |
|---|---|---|---|
| Poisson-Gaussian Noise (lam) | 10 | 25 | 75 |
| Color Shift (RGB offset) | ±30 | ±60 | ±120 |
| Resolution (scale factor) | 2× | 4× | 8× |
| Frame Drop (drop rate) | 0.1 | 0.2 | 0.3 |
| Light Flicker (frequency) | 20 | 50 | 80 |
| Rolling Shutter (ratio) | 0.1 | 0.2 | 0.5 |
| Actuator Latency (delay steps) | 5 | 15 | 25 |
| Packet Loss (drop rate) | 0.1 | 0.2 | 0.3 |

| Basic Control Primitives | |
|---|---|
| Primitive | Success Rate (%) |
| Pick Cube | 96 |
| Place Sphere | 88 |
| Press Switch | 72 |
| Push Cube | 92 |

Table 9: Success rates of the four basic control primitives.

## A.5 CONTROL-NORMALIZED ANALYSIS

Recognizing that control and other capability families are not perfectly separable, we design an experiment that explicitly factors out the impact of Control. We evaluate GR00T-1.5 on four foundational control primitives that appear across all NEBULA tasks: *Pick Cube*, *Place Sphere*, *Press Switch*, and *Push Cube*. Table 9 reports the success rates for these primitives and serves as an estimate of the agent's baseline motor proficiency.

To isolate the contribution of the target capability for family *Language*, *Dynamic*, and *Spatial Reasoning*, we normalize each capability-task success rate by the success rate of its corresponding control primitive:

$$\text{Normalized Accuracy} = \frac{\text{Task Success Rate}}{\text{Control Primitive Success Rate}}.$$

Table 9 and Figure 23 present average normalized performance across the three difficulty tiers and the radar plot of the general performance. For the *Perception* family, accuracy is evaluated based on the contact with the target object rather than specific actions, while for the *Robustness* family, as a variant of control tasks, it is standardized by its original version. After removing the control component, the resulting monotonic performance decline with increasing difficulty demonstrates that our tiered task design successfully modulates the intended capability rather than being dominated by incidental control failures.

These findings validate NEBULA's bottleneck-dominance design that once control influence is factored out, the remaining variance in performance is attributable to the capability dimension being tested.

|          | **Easy**  | **Medium** | **Hard**  |
|----------|-----------|------------|-----------|
| Language | 95.83%    | 89.58%     | 81.94%    |
| Dynamic  | 49.62%    | 43.06%     | 23.86%    |
| Spatial  | 42.70 %   | 12.50 %    | 0.00%     |

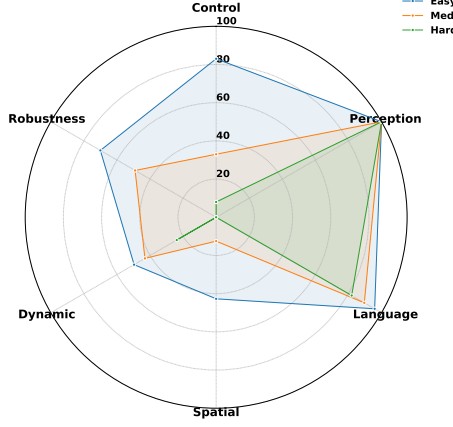

Figure 23: Normalized performance of GR00T-1.5 across the three difficulty tiers after removing control influence.

