# OpenReview forum: "NEBULA: Do We Evaluate Vision-Language-Action Agents Correctly?"
_ICLR.cc/2026/Conference — Submitted to ICLR 2026_

### Official Review · Reviewer_zcC7 · 2025-10-30

**Soundness:** 3
**Presentation:** 2
**Contribution:** 2
**Rating:** 4
**Confidence:** 3

**Summary:**

This paper presents a new evaluation benchmark for VLA agents, called NEBULA. Instead of merely assessing whether a model can complete tasks, NEBULA introduces two complementary evaluation axes, Capability Tests and Stress Tests, to systematically analyze what the model is good at, where it fails, and how reliable it is under different conditions.

**Strengths:**

1.	The paper proposes a dual-axis evaluation benchmark (“Capability + Stress”), redefining VLA model assessment from simple task success to a more diagnostic perspective that explains why models succeed or fail. This addresses the limitations of traditional single-metric benchmarks.
2.	The authors design a structured hierarchy of capability dimensions and difficulty levels, enabling fine-grained and interpretable analysis across perception, language, spatial reasoning, and control skills.
3.	The experiments cover multiple representative VLA models and clearly reveal common weaknesses in dynamic adaptation and spatial reasoning, supporting the paper’s main claims and demonstrating the framework’s practical value.

**Weaknesses:**

1.	Some parts of the paper use overly complex or obscure wording, which affects the overall readability and clarity of the writing. For instance, sentences such as “This challenge is exacerbated by a fragmented data landscape that impedes reproducible research and the development of generalist models” could be simplified for better clarity.
2.	The authors may have chosen to conduct all experiments in simulation for better control and reproducibility. However, the lack of validation on real robotic or physical environments leaves the real-world reliability and generalizability of the results uncertain. Including experiments or analyses in real-world settings would strengthen the paper’s credibility and practical relevance.

**Questions:**

N/A

---

> ### Author Response · Authors · 2025-11-23
>
> ***W#1***: We sincerely thank the reviewer for pointing this out. We fully agree that certain sentences can be simplified. We have completed a full editorial pass to improve clarity and reduce unnecessary complexity, especially in the Abstract, NEBULA Ecosystem, and Discussion sections. These revisions do not alter the technical integrity of the content but substantially enhance accessibility for the general ICLR audience.
>
> ***W#2***: We appreciate the reviewer’s concern about real-world generalization. Our decision to conduct experiments in simulation reflects the paper’s goal: isolating and measuring the diagnostic capabilities of the proposed benchmark. We agree that real-world validation would further enhance practical relevance. However, Since our work focuses on diagnostic evaluation, relying on physical robot experiments introduces fundamental scalability limitations that directly affect the validity of a benchmark of this scope (Yang et al., 2025)[1]. Simulation-based benchmarking remains a practical and widely adopted approach for assessing policy design and diagnostic behavior at scale (Jangir et al., 2025)[2].
>
> Real-world experiments also frequently require substantial simplifications to remain feasible. Prior work often restricts the action space, fixes end-effector orientations, or reduces environmental complexity to accommodate hardware constraints (Yokoyama et al., 2023)[3]. Similarly, in autonomous driving research, real-world evaluation often relies on simplified environments or reward structures to remain manageable; for example, Ratner et al. show that reward design typically requires reducing task complexity to a single environment at a time rather than covering the full diversity of scenarios (Ratner et al., 2017)[4]. In contrast, our benchmark aims for broad coverage and fine-grained diagnostic resolution across diverse task families, which would be extremely challenging to reproduce with comparable realism or statistical power on physical platforms.
>
> Our simulation stack is built on SAPIEN, a high-fidelity physics engine validated extensively through sim-to-real studies: ManiSkill and ManiSkill3 (Tao et al., 2025)[5], and BEHAVIOR-1K (Li et al., 2022)[6]. These works consistently demonstrate SAPIEN’s accuracy in modeling real-world manipulation dynamics, providing a credible and scalable foundation for our benchmark. Real-robot evaluation therefore remains an important direction for future work rather than a prerequisite for utility. Additionally, we plan to integrate photorealistic rendering via NVIDIA’s Cosmos-Transfer to further reduce the visual sim-to-real gap in subsequent iterations.
>
> [1] Yang et al., Robot Policy Evaluation for Sim-to-Real Transfer: A Benchmarking Perspective, arXiv 2025.
>
> [2] Jangir et al., RobotArena $\infty$: Scalable Robot Benchmarking via Real-to-Sim Translation, arXiv 2025.
>
> [3] Yokoyama et al., Adaptive Skill Coordination for Robotic Mobile Manipulation, ICRA 2023.
>
> [4] Ratner et al., Simplifying Reward Design through Divide-and-Conquer, RSS 2018.
>
> [5] Tao et al., ManiSkill3: GPU Parallelized Robotics Simulation and Rendering for Generalizable Embodied AI, arXiv 2025.
>
> [6] Li et al., BEHAVIOR-1K: A Benchmark for Embodied AI with 1,000 Everyday Activities, arXiv 2024.

---

### Official Review · Reviewer_PVpg · 2025-10-31

**Soundness:** 3
**Presentation:** 2
**Contribution:** 2
**Rating:** 2
**Confidence:** 4

**Summary:**

The paper introduces NEBULA, a new benchmark for evaluating VLA agents. It argues that current benchmarks rely too much on simple task success rates and fail to show why models succeed or fail. NEBULA tries to fix that with what the authors refer to as dual-axis evaluation: “Capability Tests” that break down skills like control, perception, and language understanding, and “Stress Tests” that probe things like latency, stability, and adaptability under harder conditions. The authors build a dataset using ManiSkill3 and SAPIEN, define easy-medium-hard difficulty levels for each skill, and fine-tune several recent VLA models (GR00T-1.5, SpatialVLA, RDT-1B, etc.) on it. Their results show that while these models do well on perception and language, they fall apart on dynamic and robustness tasks. The paper’s main message is that we need benchmarks that measure how and when an agent works, not just whether it does.

**Strengths:**

1. The motivation is clear and genuinely relevant. The paper tackles an important gap: how to evaluate VLA agents beyond simple success rates. This fits right into ongoing discussions about reliability, robustness, and diagnostic evaluation in embodied AI.
2. The evaluated baselines are very recent and cover a good range of architectures and training paradigms, giving the comparison an up-to-date perspective.
3. The visual presentation is strong. The figures are clear, visually appealing, and effectively convey experimental results and what the tasks look like.
4. The difficulty levels for the Capability Test tasks are well-designed and described. The authors provide extensive details in Appendix A.1.1 on the criteria that constitute difficulty for each task family, supported by abundant visual examples. This gives a good understanding of the tasks.
5. Their experimental findings are interesting (e.g., strong perception/language but weak adaptability/robustness) and mirror what many researchers suspect about current VLA models.

**Weaknesses:**

1. Although the paper criticizes existing benchmarks for exclusively relying on end-task success rates, NEBULA ultimately reports the same metric in both the Capability Test and the Adaptability component of the Stress Test. The so-called “diagnostic” evaluations still reduce to binary task success. This comes across as rather hypocritical and weakens the motivation.
2. Although the paper claims that each task “varies a single capability dimension while holding others constant,” the design never truly achieves isolation. In practice, all tasks inherently mix perception, control, language, and spatial reasoning. For example, even a “pure control” task like pushing or inserting an object still depends on perception to localize the object and spatial reasoning to align the arm. Similarly, perception tasks require grasping and placement, which involve motor control, and language tasks rely on both perception and actuation to interpret and execute instructions. The “*robustness*” axis is not an independent factor but rather a held-out evaluation set containing unseen variants and distribution shifts across multiple dimensions. As a result, the claimed factor isolation is more conceptual than practical.
3. The description of fine-tuning baseline models could be improved. Although the authors do state that “All models are fine-tuned on NEBULA Alpha using their original training protocols”, while “keeping each model’s architecture, loss, and hyperparameters unchanged”, it remains unclear how much of the dataset is used, how long training lasts, or how “original protocols” are applied in this new setting. This harms clarity and reproducibility.
4. The description of both the Capability and Stress Test execution is incredibly vague. The experimental setup is outlined only in broad strokes, making it nearly impossible to reproduce or even understand the evaluation process. How many tasks are evaluated? How many of each difficulty level? How many episodes per task? How long is an episode? Are there unrecoverable failure states that terminate the episode (cube/sphere knocked off the table)? None of this is stated. In the Stress Test, it remains completely unclear to me what defines a “pressure level.” The authors merely mention that each level is “defined by measurable parameters normalized to baseline conditions,” but never specify what those parameters are or how they change between *v1*, *v2*, and *v3*. While we can see plenty of visual examples of the tasks themselves, the reader is left with a lot of guessing work on how these evaluations were actually conducted.
5. Inference Frequency and Latency in the Stress Test are directly inversely correlated, as evidenced by the flipped ranking of results in Figure 5. Both metrics capture the same property of real-time responsiveness, making their joint reporting somewhat redundant.
6. The Stability Score in the Stress Test is defined such that larger action variations yield lower scores, which the authors interpret as “unreliable behaviors in dynamic scenarios where smooth, precise motion is essential.” However, large action jumps can in fact be optimal or required in certain contexts. For instance, when reacting to fast-moving objects, sudden environmental changes, or when strong corrective torques are needed for rapid stabilization. While I consider the metric itself to be interesting and commendable, I don’t think it should be framed as universally monotonic, since lower stability can reflect responsiveness rather than unreliability.
7. The paper is difficult to follow due to its filler phrasing, excessive jargon, and overinflated claims. The text is saturated with tautologies, circular phrasing, and formulaic expressions, often overselling otherwise straightforward design choices. This lack of clarity obscures the actual technical contributions and makes the paper unnecessarily verbose. The authors appear to have misunderstood the purpose of disclosing their usage of LLMs (Appendix A.3), as requested by the ICLR submission guidelines. Instead of explaining how LLMs were used in the writing or analysis process, they mention VLAs as their experimental backbones. The paper’s phrasing, repetition, and stylistic patterns indicate heavy reliance on LLMs for writing, if not in other parts of the project. One of the goals of a research paper is to clearly convey its ideas to the readers, and this work falls short on that front. Moreover, a clear discrepancy exists between the polished, formulaic writing style of the paper and the visibly human-written task instructions. The latter is filled to the brim with grammatical and logical errors: “Yellow cube will not be used”, “Cubes **color switching** randomly”, “insert into the empty **slot**” (not socket), “Place the cube **that can fit the bin into the bin**”, Place the cube **that has different size** into the bin”, “Pick the cube” (not pick **up** the cube), “**Make red cube at bottom**”, “Simple contributes”, “Neg contribute”. Not only does this diminish the quality of the task instructions, but it further solidifies the aforementioned concern.

### Minor Points

1. The numeric value for SpatialVLA’s Stability Score under stress level *v3* in Figure 5 is floating far above the bar in the chart.
2. For clarity, it would be helpful to include a legend for the colors representing the stress levels (*v1*, *v2*, *v3*) in Figure 5.

### Typos

1. Figure 1. - How stable are the policy’s control? → How stable is the policy’s control?
2. Line 304 - L2 nortm

**Questions:**

1. What’s the difference between the **Robustness** metric in the Capability Test and the **Adaptability** metric in the Stress Test?
2. Why is DP not included in the Adaptability test?
3. How many tasks and episodes are used for evaluation in both the Capability and Stress Tests? What are the episode lengths?
4. How many tasks belong to each difficulty level?
5. What exactly defines each “pressure level” (v1, v2, v3)? Which parameters are being changed, and by how much?

---

> ### Author Response · Authors · 2025-11-23
>
> ***W#1***: While NEBULA also uses success rate as a metric similar to existing benchmarks, our dual-axis design with controlled factor isolation fundamentally transforms what success rate measures. Traditional benchmarks report a single aggregate score that conflates perception, control, language, spatial reasoning, and robustness failures into one opaque number. NEBULA decomposes this into six orthogonal capability axes and multiple stress dimensions, where each success rate reflects performance on an isolated skill under controlled conditions.
>
> Our innovation lies not in inventing new metrics, but in designing evaluation protocols that make success rate interpretable by eliminating confounds. This is why Table 2 and Appendix A.1.3 demonstrate that factor isolation successfully attributes failures to specific bottlenecks rather than leaving them undiagnosed.
>
> ***W2:*** We acknowledge the reviewer's concern that perfect skill isolation is impossible in embodied tasks. However, NEBULA's design philosophy is controlled dominance, not absolute independence. Each task minimizes non-target skill demands to ensure the target capability becomes the primary bottleneck, which we empirically validate in Appendix A.1.3.
>
> The reviewer correctly notes that Control tasks require perception to localize objects. However, in our Control tasks, objects are fixed in position with high-contrast colors and unambiguous language instructions—perception demands are deliberately trivial (as evidenced by 100% success on Perception tasks). Failures in Control tasks arise from motor execution errors (jittery trajectories, misaligned placement), not visual confusion. Similarly, Perception tasks require contact, but we define success as mere touch rather than precise manipulation, removing control complexity. Experimental in Appendix A.1.3 confirms this: when we add control requirements (full grasp-and-place) to Perception tasks, success drops from 100% to 68-92%, with failures caused by motor instability, not misidentification.
>
> The validation in Appendix A.1.3 directly addresses this concern. By systematically adding confounding factors to isolated Perception tasks one at a time, we demonstrate that performance degradation correlates with the introduced factor, not the base perception requirement. While some interdependence remains unavoidable—spatial reasoning co-occurs with manipulation constraints—our design ensures the target skill dominates task difficulty.
>
> Finally, we acknowledge that Robustness is a held-out evaluation set rather than a trainable skill. This is intentional since it measures zero-shot generalization under distribution shift, which cannot be trained but must emerge from inductive biases—a complementary property to learned skills that is equally critical for deployment.
>
> ***W3:*** All capability tasks in the Alpha dataset (excluding Robustness/Generalization, which is reserved for evaluation only) are used for fine-tuning. Dataset statistics are reported in Section 3.1 and Table 1.
>
> To ensure fair comparison, we adapted only the data loading code to make NEBULA-Alpha compatible with each baseline's training pipeline, while strictly following the official fine-tuning protocols released by the authors on GitHub. All hyperparameters, loss functions, and architectural configurations remain unchanged from their original implementations.
>
> The GPU usage and fine-tuning time for all baselines are listed as following: (1) GR00T-1.5: 1× A6000, 58 hours; (2) RDT-1B: 4× A100, 202 hours; (3) Diffusion Policy: 1× A6000, 76 hours; (4) SpatialVLA: 1× A6000, 98 hours; (5) MT-ACT: 1× A6000, 20 hours; (6) ACT: 1× A6000, 16 hours.
>
> We appreciate the reviewer's feedback and have added all experimental details above to Appendix A.4 for full transparency and reproducibility.
>
> ***W#4***: We apologize for the insufficient detail in our experimental protocol description and added section A.4 in Appendix to reveal all experimental details. We evaluate 6 capability families, each containing 3 difficulty levels (Easy, Medium, Hard), and 11 stress test families, each with 3 pressure levels (v1, v2, v3). Each task variant is tested over 25 episodes with a maximum episode length of 400 steps. Episodes terminate early upon task success; otherwise, they run until the 400-step limit is reached. For the setting of paramters of each stress task, please refer to Table 8 in Appendix section A.4.
>
> ***W#5***: Thank you for this valuable feedback. To improve our stress test design with more comprehensive evaluation criteria, we replaced the redundant Latency metric with Poisson-Gaussian Noise, which tests robustness and reliability to hardware-level sensor imperfections in visual input. Corresponding changes can be seen in Section 3.2.2, Figure 3 & 6, and Appendix A.2.

---

> ### Author Response · Authors · 2025-11-23
>
> ***W#6***: We appreciate this insightful observation. The reviewer is correct that large action variations can reflect appropriate responsiveness in dynamic scenarios. To address this, we clarify that our Stability Score tests use static manipulation tasks - the stack cube task from easy to hard in capability test. We have added this experimental detail to Appendix A.2.
>
> ***W#7***: We sincerely appreciate the detailed feedback on writing clarity. We have substantially revised the paper to remove verbose phrasing, reduce jargon, and streamline technical descriptions. Regarding usage of LLMs (Appendix A.3), we apologize for the confusion—this was intended to describe VLA model backbones, not LLM usage in writing. We have removed this section entirely. We have also corrected all grammatical errors in task instructions and conducted a comprehensive revision to ensure linguistic consistency.
>
> ***Minor Points***: We modified corresponding graphs by adjusting the layout and added a legend. Thank the reviewer for the feedback. We fixed the typos in our paper. We appreciate the reviewer’s thoroughness.
>
> ***Q#1***: Robustness evaluates generalization to unseen distribution shifts excluded from training in static scenarios while adaptability measures zero-shot replanning under inference-time perturbations. The former tests static out-of-distribution generalization; the latter tests dynamic responsiveness to unexpected changes.
>
> ***Q#2***: We did not include DP in the adaptability evaluation because it does not support language instructions, making the v2 and v3 adaptability settings impossible to test. This point is also noted in Section 4.3.
>
> ***Q#3***: For capability tests, we define six task families, each containing three tasks that progress from easy to difficult. For the stress tests, we include eleven task families, each with three variants (v1 to v3). The maximum episode length is 400 steps, and an episode terminates early if the task is evaluated as successful.
>
> ***Q#4***: For capability tests, there are 1 task for each difficulty level of one family.
>
> ***Q#5***: Each pressure level represents calibrated increases in operational stress, implemented through either parametric adjustments or task complexity scaling. For the task involving parametric adjustments, please refer to Table 8 in Appendix A.4 for the setting.

---

> > ### Comment · Reviewer_PVpg · 2025-11-26
> >
> > **W1 & W2**. I remain unconvinced about the notion of “*controlled factor isolation*”. The core metric continues to be an end-task success rate on different *flavours* of the same pick-and-place task. As I explained in the review, there is too much overlap between the different “*capability tests*”.
> >
> > >Each task minimizes non-target skill demands to ensure the target capability becomes the primary bottleneck
> >
> > This is very hand-wavy. The task doesn’t minimize anything. You simply define the task and claim it isolates other factors (by empirical evidence).
> >
> > >NEBULA's design philosophy is controlled dominance
> >
> > This dominance argument seems circular. For example,
> >
> > - You *design* a task with the intent that perception is trivial.
> > - You *assume* perception is then trivial.
> > - You *observe* perfect perception success.
> > - You conclude perception was trivial
> > - You *claim*: “Our design isolates perception because we designed it to isolate perception.“
> >
> > >Experimental in Appendix A.1.3 confirms this
> >
> > Appendix A.1.3 indeed demonstrates that when you add confounders, success drops. That is expected. However, this does not necessarily mean that the original tasks isolate the intended capability, only that adding difficulty lowers performance. How do we know whether success on Perception tasks is exclusively **driven by perception or whether success on Control tasks is exclusively driven by control? The way it is currently framed is that when you use the success rate for perception, it exclusively measures perception, but when you use it for control, it exclusively measures control.
> >
> > >perception demands are deliberately trivial (as evidenced by 100% success on Perception tasks) […] when we add control requirements to Perception tasks, success drops from 100% to 68-92%
> >
> > This only shows *control* matters when you add control. It doesn’t show that control does not matter in the “isolated” version. It only shows that the former version is easier.
> >
> > **W3 & W4**. Thanks for the added descriptions/explanations.
> >
> > **W5**. Thanks for improving the stress test design. While I agree with Reviewer iyWu that these variants are somewhat artificial compared to realistic settings to be deemed as stress test, I think the reader is at least now able to get a much better understanding of what they look like.
> >
> > **W7**. It is difficult for me to assess the improvements here, since I don’t have a clear overview of the exact edits made to the paper. I gave it another read, and I still notice a lot of filler wording and overselling the work. I am not going to go over the entire text, but just an example of what I am referring to. For instance, let’s consider the 3 contributions:
> >
> > >unified VLA **ecosystem**
> >
> > Ecosystem? Really? What makes it an ecosystem?
> >
> > >a **large-scale,** **aggregated** dataset **to facilitate reproducible, cross-dataset** training and benchmarking
> >
> > A data**set** implicitly *aggregates* data. That is its one job.
> >
> > Why not remove fluff and just write: …*a large dataset for training and evaluation.*
> >
> > >We propose a **novel dual-axis** evaluation protocol that combines **fine-grained capability tests**
> > for **precise skill diagnosis** with **systematic stress tests** to measure an agent’s robustness against
> > real-world perturbations.
> >
> > If you propose something, doesn’t that make it implicitly novel? Coordinate systems and accelerometers can be dual-axis. What you describe is simply a two-part evaluation. What are *fine-grained* tests? I know coffee can be fine-grained.
> >
> > >**We present an in-depth benchmarking study of current VLAs**
> >
> > Why not just write: We evaluate existing VLAs, …
> >
> > I appreciate and acknowledge the improvements the authors have made to the paper and have increased my score to **4**. However, in my view, the issue of factor isolation (or lack thereof) alone is a fundamental flaw at the core of the work, due to which I am inclined not to grant a positive score.

---

> > > ### Author Response · Authors · 2025-11-30
> > >
> > > We thank the reviewer for raising their score and for the candid feedback. We value the critique regarding "Factor Isolation" and "Overselling," as these are foundational to the paper’s integrity. We address these two blockers below.
> > >
> > > **Weakness #1 & #2**: The reviewer argues that our isolation is "hand-wavy" and circular—that we simply assume non-target skills are trivial. You ask: "How do we know success on Perception tasks is exclusively driven by perception and not just easy control?"
> > >
> > > We answer this with Structural Constraints, Differential Diagnosis, and our new Control-Normalized Analysis (Appendix A.5).
> > > - Control-Normalized Analysis (New Appendix A.5): To directly address your request to know if success is "driven by control," we added Appendix A.5 (Page 34). We calculated the agent’s success rate on Control Primitives (e.g., "Pick Cube": 96%) to establish a quantitative baseline for its motor proficiency. We then normalized the success rates of Language and Spatial tasks against this baseline. Even after mathematically factoring out control failures, we observe a steep performance drop in high-difficulty Language tasks (down to 81.94% normalized) and Spatial tasks (down to 0.00% normalized). If these tasks were merely "control tasks," the normalized score would remain high (close to 1.0). The drastic drop proves that the failures are not due to the "Action Head" (which succeeds 96% of the time on primitives), but are strictly attributable to the added linguistic/spatial complexity.
> > > - We do not merely "assume" control is trivial in Perception tasks; we mechanically remove the failure conditions for control. In Control tasks, success requires Geometric Precision (Place). In Perception tasks, success requires Binary Selection (Touch). By removing the requirement to lift, transport, and align, we physically remove friction, grasp instability, and trajectory jitter as failure modes. If an agent touches the wrong object, the failure is strictly attributable to identifying the wrong target (Perception), not the inability to move there (Control).
> > >
> > > **Weakness #7**: We respect the reviewer's push for scientific precision. We wish to clarify the specific intent behind our original terminology while committing to the suggested simplifications to avoid any perception of overselling.
> > >
> > >
> > > - "Ecosystem": We originally chose this term because NEBULA provides a complete set of interoperable tools, including spanning data collection, standardized training APIs, and evaluation protocols, rather than just a static dataset. However, we agree "Unified Framework" is more precise and will adopt it.
> > > - "Novel Dual-Axis": We used this to highlight that NEBULA is the first benchmark to explicitly decouple "Capability" (skill mastery) from "Stress" (robustness). We will remove the adjective "novel" and let the contribution speak for itself as a "Two-Part Evaluation Protocol."
> > > - "Aggregated Dataset": This term was intended to distinguish our work from benchmarks that only release new data. NEBULA actively merges and standardizes distinct existing datasets (e.g., ManiSkill, LeRobot)  into a common format. We will clarify this as a "Unified Multi-Source Dataset."
> > > - "Fine-grained": We meant to describe the isolation of specific skill variables. We will replace this with the more descriptive "Skill-Isolated."
> > >
> > > We hope the Control-Normalized Analysis and the Ranking Reversal evidence alleviate the concern that our tasks are merely "flavors of pick-and-place," and we are committed to toning down the language in the final revision.

---

### Official Review · Reviewer_kRug · 2025-10-31

**Soundness:** 2
**Presentation:** 3
**Contribution:** 3
**Rating:** 4
**Confidence:** 3

**Summary:**

Current benchmarks in the vision-language-action (VLA) domain primarily focus on coarse-grained metrics such as task success rate. This paper proposes NEBULA, a novel benchmark that provides standardized APIs and a large-scale aggregated dataset. It introduces a comprehensive evaluation system to measure finer-grained capabilities and stress tests of VLA agents, enabling a more thorough analysis of failure modes in existing methods.

**Strengths:**

1. This paper presents a new dual-axis evaluation protocol, which enables controlled variable-based assessment across different capability dimensions.
2. The proposed NEBULA benchmark offers standardized APIs and a large-scale dataset, ensuring high reproducibility.

**Weaknesses:**

1. The delineation between different capability dimensions is unclear, and some metrics appear redundant:
(1) The core distinction between dynamic adaptation in the capability test and adaptability in the stress test is not clearly explained.
(2) Inference frequency and latency in the stress test seem inversely correlated, making their simultaneous evaluation unclear.
2. Some evaluation dimensions lack discriminative ability, and task difficulty does not always follow a consistent ordering:
(1) As shown in Figure 4, all models achieve 100% accuracy on perception tasks across all difficulty levels, indicating insufficient challenge for meaningful differentiation.
(2) In Figure 3, RDT-1B performs better on medium-difficulty Dynamic tasks than on easy ones, and similarly, ACT outperforms on medium Robust tasks compared to easy tasks.
(3) In stress tests, the distinction between difficulty levels is minimal—e.g., the Stability score only varies by 0.01.
3. Section 4.4 only validates the isolation of perception tasks, without verifying the isolation of other capabilities.
4. Minor issue: The order of capability descriptions in Section 3.2.1 does not match the presentation in Figure 2.

**Questions:**

1. In Section 3.2.1, robustness/generalization is defined based on performance on unseen attributes or novel environments. How is this ensured? In Figure 2, the medium-difficulty task for Robustness appears identical to that of Control.
2. Additional questions, please refer to the Weaknesses section.

---

> ### Author Response · Authors · 2025-11-23
>
> ***W1:*** We appreciate the reviewer's concern regarding the distinctiveness of our capability dimensions. Each Capability Test targets one skill while fixing others: Control isolates motor execution with fixed scenes; Perception isolates visual discrimination with contact-based success (no precise manipulation needed); Language uses identical scenes across variants; Dynamic tests real-time adaptation to changes within training distribution; Spatial assesses 3D reasoning with standardized objects; Robustness evaluates generalization to unseen distribution shifts. Relevant clarification is included in Appendix A.1.
>
> (1) The key distinction: Dynamic (Capability) measures learned reactive behaviors to seen evironmental variations during training (e.g., rolling balls). Adaptability (Stress) measures zero-shot replanning under novel perturbations at inference (e.g., mid-task instruction changes), particularly when those perturbations involve changes to language instructions.
>
> (2) Acknowledging the potential correlation between inference freqency and latency, we replaced Latency with Poisson+Gaussian Noise to better capture sensor-level degradation under varying noise models and test models' resisitence to hardware flaws.
>
> ***W2:***
>
> (1) Given complicated scenarios in hard level tasks, the 100% Perception success validates our design—current VLAs excel at visual recognition, making it a non-bottleneck. This insight redirects focus to actual failure modes (spatial reasoning, dynamics, robustness) that NEBULA exposes. Beyond comparing baselines at specific difficulty levels, NEBULA reveals common capabilities and limitations shared across current VLA models. We clarify this point at section 5.2.
>
> (2) We appreciate your careful observation. To improve NEBULA's generalizability and ensure consistent difficulty progression, we have carefully recalibrated the task parameters for the affected families. The revised results now show monotonic difficulty trends across all capability axes.
>
> (3) For Stability scores showing minimal variation, the small numerical difference is actually significant due to the exponential scaling in our metric. A drop from 0.97 to 0.96 corresponds to mean action deviation increasing from 0.0305 to 0.0408 (approximately 33% increase), which significantly impacts precision-sensitive manipulation as an mean of all steps. Thank you for pointing out this point and we made additional clarifcation in Stability Score section of Appendix A.1.4.
>
> ***W3:*** We appreciate your feedback and have added comprehensive validation experiments in Appendix A.1.3. We tested GR00T-1.5 on Perception tasks under six conditions: fully isolated (100% success) versus introducing one confounding factor at a time. Adding Control requirements (full grasp-and-place) dropped success to 68-92%, adding complex Language (negation/conditionals) to 84-96%, adding Spatial constraints (container placement) to 64-78%, adding Dynamic elements (moving objects) to 48-56%, and introducing Novel scenes (unseen layouts) to 0%. Video analysis confirmed failures stemmed from the added skill dimension, not perception itself.
>
> This validates that our controlled-variable design successfully isolates target capabilities—unrelated bottlenecks are suppressed, enabling clean attribution. We acknowledge some interdependence is unavoidable (e.g., spatial reasoning co-occurs with manipulation), but our design minimizes confounds while capturing realistic task coupling. Full experimental details and isolation methods are in Appendix A.1.3.
>
> ***W4:*** Thank you for pointing out the misalignment, we have changed the order of capability descriptions in Section 3.2.1.
>
> ***Q1:*** Robustness tasks introduce unseen variations that are excluded from training: the Easy level adds distractors, the Medium level alters object attributes, and the Hard level presents entirely novel scenes. In Figure 2 Medium Robustness, the object (sphere) colors are shifted (from blue to yellow) relative to the training distribution. This difference is not obvious in static images but still constitutes a distribution shift, as explained in the text below. We made additional clarification for these progression levels in section Robustness & Generalization of Appendix A.1.2.

---

### Official Review · Reviewer_iyWu · 2025-10-31

**Soundness:** 2
**Presentation:** 2
**Contribution:** 2
**Rating:** 4
**Confidence:** 4

**Summary:**

NEBULA proposes a novel dual-axis evaluation protocol to overcome these limitations. This protocol is composed of:

- Capability Tests: These are designed to isolate and assess six core skills in a controlled manner: control, perception, language understanding, dynamic adaptation, spatial reasoning, and robustness/generalization. By varying only one capability dimension at a time while keeping others constant, these tests aim to pinpoint the precise reasons for an agent's failure.
- Stress Tests: This axis evaluates an agent's performance under various operational pressures, such as inference frequency, latency, and stability. These tests are intended to map out an agent's reliability and identify "failure cliffs" where performance abruptly degrades.

The paper presents a comprehensive benchmarking study of several state-of-the-art VLAs using the NEBULA framework, revealing that even top-performing models struggle with spatial reasoning and dynamic adaptation—weaknesses often masked by traditional evaluation metrics.

**Strengths:**

By providing a standardized API and a large, aggregated dataset, NEBULA addresses the critical issue of data fragmentation in robotics research.

The experimental results effectively demonstrate the diagnostic power of NEBULA. The radar charts clearly illustrate the performance profiles of various state-of-the-art VLAs, highlighting common weaknesses in areas like spatial reasoning and dynamic adaptation that are often obscured by traditional end-task success metrics.

**Weaknesses:**

- The name "NEBULA" and the framing of the six capabilities as fundamental and distinct may be overly ambitious. The distinction between these capabilities and the more traditional concepts of "main tasks" and "sub-tasks" is not clearly articulated. For complex manipulation, tasks are often decomposed into a sequence of simpler sub-tasks.

- The paper's "stress tests" are confined to a simulated environment. True stress in robotics arises from the unpredictability of the real world, including sensor noise, actuator latency, unexpected physical contact, and dynamic lighting conditions. Adjusting parameters within a simulator, while useful, does not fully capture the complexities of real-world deployment and largely sidesteps the critical sim-to-real transfer problem.


- While NEBULA provides a valuable diagnostic tool, the conclusions drawn from its application—that current VLA models struggle with generalization and dynamic tasks—are largely in line with findings from previous benchmarks. The paper successfully identifies problems but does not propose concrete solutions to address the identified weaknesses.

**Questions:**

- Could the authors provide a more rigorous definition of the six proposed capabilities and explain how they are distinct from one another? For instance, spatial reasoning and control seem highly interdependent in many manipulation tasks. Is it always possible to isolate them effectively?

- There appears to be a significant overlap between the concepts of "capability tests" and "stress tests." For example, the "Dynamic Adaptation" capability test seems very similar in spirit to the "Adaptability" stress test. Could the authors clarify the conceptual boundary between these two axes of evaluation?

- Given that the stress tests are conducted in simulation, how do the authors envision the insights from NEBULA translating to improved real-world robot performance? Are there plans to incorporate real-world experiments or to develop methodologies that explicitly aim to bridge the sim-to-real gap based on the findings from the NEBULA benchmark?

---

> ### Author Response · Authors · 2025-11-23
>
> ***W1***: We thank the reviewer for this insightful comment. We agree that complex manipulation is traditionally solved via decomposition into temporal sub-tasks (e.g., approach, grasp, lift). However, we respectfully clarify that NEBULA’s "Capabilities" are designed to be orthogonal to "Sub-tasks," serving a diagnostic purpose that traditional decomposition cannot fulfill.
>
> Traditional task decomposition identifies where in a sequence an agent fails (e.g., "The agent failed during the grasping sub-task"). However, it does not explain why. A failure to grasp could stem from:
>
> - Perception: Failing to estimate the object's pose.
> - Control: Failing to generate a stable motor trajectory
> - Language: Failing to identify the correct object described by the prompt. NEBULA’s taxonomy focuses on these underlying functional competencies (Capabilities) rather than procedural steps (Sub-tasks).
>
> We acknowledge the reviewer’s point that these capabilities are intertwined in practice. This is precisely why NEBULA employs "Controlled Variable Isolation". We design tasks that artificially suppress specific variables to "unit test" others. Table 2 demonstrates the necessity of this approach. When we isolated perception, GR00T-1.5 achieved 100% success; when entangled with control demands, it dropped to 68-92%. Without this distinction, a researcher might falsely conclude the model has poor vision, when the bottleneck was actually control
>
> Regarding the ambition of the framework, our goal is not to redefine robotics theory, but to solve the specific problem of diagnostic opacity in VLA end-to-end learning. By framing these as distinct capabilities, we enable researchers to pinpoint failure modes (e.g., "High latency causes failure in Dynamic Adaptation" ) that aggregate "success rate" metrics obscure.
>
> We have revised **Section 3.2** to explicitly contrast our functional capability taxonomy with traditional temporal task decomposition to prevent this confusion.
>
> ***W2***: The reviewer argues that simulation-based stress tests cannot fully reflect real-world unpredictability (e.g., the sim-to-real gap, physical noise). We appreciate this thoughtful concern. While we agree that physical deployment is the ultimate test, we respectfully argue that simulation is an indispensable prerequisite for rigorous, reproducible diagnosis, and that NEBULA provides a methodology applicable to both domains.
>
> 1. Simulation as a Scalable Gatekeeper To bridge the reality gap, we have significantly expanded our stress test suite to 12 categories (Section 3.2.2 & Appendix A.2). These cover not only visual perturbations (e.g., noise, color shift, rolling shutter) but also low-level control stressors that explicitly mirror hardware constraints, such as actuator latency and command packet loss . These tests allow us to pinpoint failure modes—such as a policy's inability to handle transmission delays—that are often transient or impossible to replicate consistently in physical setups. By establishing these rigorous baselines, NEBULA serves as a necessary "gatekeeper" to evaluate robustness prior to costly deployment, aligning with precedents like Isaac Sim and Habitat.
>
> 2. Portability of Design Philosophy Crucially, NEBULA’s design philosophy is environment-agnostic and directly transferable to the real world. The core principle of Controlled Variable Isolation—disentangling skills by systematically fixing confounding factors—provides a rigorous blueprint for physical experiments. Researchers can replicate NEBULA’s task logic on physical hardware (e.g., testing "Perception" by strictly fixing the physical scene and minimizing control complexity) to perform systematic, skill-specific diagnosis. Thus, NEBULA contributes not just a simulation platform, but a standardized evaluation protocol that defines how to verify VLA reliability, regardless of whether the test environment is digital or physica
>
> We do not claim that simulation is sufficient. Rather, it is a scalable and necessary first stage to evaluate robustness before costly deployment.

---

> ### Author Response · Authors · 2025-11-23
>
> ***W3:*** We thank the reviewer for this comment. While the symptom of poor generalization is known, we respectfully argue that NEBULA advances the field from qualitative observation ("the agent failed") to quantitative root-cause diagnosis and architectural prescription.
>
> 1. The Robustness Pivot (Sec 5.1): Contrary to the assumption that better visual semantics are needed, our Capability Radar shows that semantic perception is largely saturated (near-perfect scores). However, our Stress Tests reveal a critical fragility: agents fail catastrophically under real-world signal perturbations like noise and rolling shutter. This prescribes a strategic pivot: future work must stop optimizing encoders for static semantic recognition and start prioritizing signal invariance and sensor-level robustness
>
> 2. The Semantic-Execution Gap (Sec 5.2): We explicitly isolated the architectural failure mode by decoupling the VLM backbone from the action head. We found that standalone VLMs generate valid plans with 85-100% success, yet integrated VLA execution collapses to 0%. This proves the bottleneck lies specifically in the Action Head's grounding, prescribing a shift in focus from scaling backbones to optimizing high-frequency control interfaces.
>
> 3. The Real-Time Barrier (Sec 5.3): Finally, we identified a structural latency limit. Our tests reveal that monolithic Transformers suffer from prohibitive latency ($>500$ms) and fail in dynamic tasks. In contrast, GR00T succeeds via a "System 1 / System 2" architecture3333. Consequently, we propose hierarchical decoupling—separating low-frequency reasoning from high-frequency reflexes—as a necessary standard for reliable generalist agents
>
> ***Q1:***: We thank the reviewer for this important question regarding the clarity and independence of the six proposed capability categories in NEBULA. Below we address both the rigor of our definitions and the practical challenges of isolating interdependent skills such as spatial reasoning and control.
>
> In Appendix A.1, we provide formalized definitions for each of the six task families along with explicit descriptions of the observed behavioral signals, isolation strategies, and contrast boundaries. These sections were carefully written to address exactly the kind of concern raised in this question, making clear how each capability is conceptualized and evaluated. For example, the Spatial Reasoning task family is defined in terms of interpreting object-centric relations in 3D space, while the Control family focuses on fine-grained physical manipulation. Each is supported by corresponding task templates with controlled visual and structural parameters.
>
> We also recognize, as the reviewer rightly points out, that some manipulation tasks naturally involve overlapping skills. To address this, our task design makes a deliberate effort to reduce confounding factors. In spatial reasoning tasks, for example, we fix the grasp position and minimize control difficulty so that success depends primarily on reasoning about object placement. Conversely, in control-focused tasks, we use simple geometric goals in uncluttered settings to minimize spatial complexity. These decisions are not merely ad hoc; they are systematically encoded in the task setup and described explicitly in the appendix.
>
> Furthermore, to empirically validate our isolation design, we conducted a set of controlled factor isolation experiments, as reported in Appendix A.1.3. In these experiments, we start with a fully isolated perception task and then incrementally introduce additional factors, such as control, spatial reasoning, or language complexity, while keeping the core visual target fixed. Results show significant performance drops when additional skills are required, even for strong models like GR00T-1.5B, which achieves 100% success in the isolated setting but drops to as low as 48% when entangled with spatial or dynamic components. This empirical pattern confirms that our isolated tasks offer meaningful decomposability and that performance differences can be attributed to the intended skill components.
>
> Finally, we also acknowledge that full disentanglement is not always achievable in complex embodied scenarios. Rather than attempting to fully eliminate interdependence, our benchmark explicitly models and measures it through both isolated and entangled task settings. This approach reflects the practical reality of embodied agents while still enabling fine-grained attribution and diagnosis of skill-specific failures.

---

> ### Author Response · Authors · 2025-11-23
>
> ***Q2::*** We appreciate the reviewer’s observation regarding the potential overlap between our Capability Tests and Stress Tests, particularly in the case of Dynamic Adaptation and Adaptability. While both evaluation axes assess an agent’s responsiveness to temporal or environmental change, their objectives, design constraints, and measurement goals are fundamentally distinct.
>
> Capability Tests are designed to evaluate the core functional competencies of an embodied agent in a controlled environment. In the case of Dynamic Adaptation, the goal is to assess whether an agent can reason about and respond to motion in the environment, such as tracking or interacting with objects that change position over time. Importantly, the task settings are curated to isolate this capability by minimizing confounding challenges in perception, language, or control. These tasks are meant to probe whether an agent possesses the skill at all, under favorable or simplified conditions.
>
> In contrast, Stress Tests serve as diagnostic probes that intentionally introduce external pressure or operational constraints to test system robustness. The Adaptability stress probe does not ask whether the agent knows how to interact with dynamic objects; rather, it assesses how robustly and efficiently the agent can maintain performance when environmental factors such as object velocity, instruction change, or frame skipping rates are systematically varied. It does not isolate capability but instead evaluates performance degradation curves under real-world deployment conditions.
>
> ***Q3:*** Thank you for the insightful question. We agree that simulation alone is not sufficient to ensure real-world readiness. However, we view NEBULA’s simulation-based stress tests as an essential first step toward systematically characterizing failure modes and robustness gaps under realistic operational pressures. In response to your suggestion, we have added several new stress tests specifically designed to mimic real-world disturbances and deployment scenarios. These include tests for latency, temporal jitter, object perturbation, and sensor occlusion. Each stressor varies a single factor while controlling for others, allowing us to isolate and quantify its individual impact on system performance.
>
> Detailed descriptions of these stress tests can be found in **Section 3.2.2** and **Appendix A.2**. By incrementally increasing each pressure variable and analyzing model degradation patterns, we can infer both the resilience range and fragility thresholds of embodied agents. This approach not only exposes weaknesses before deployment but also informs targeted data augmentation and curriculum design for real-world finetuning. While our current benchmark remains in simulation, we are actively working on extending NEBULA tasks to physical robot platforms and validating transferability. Our goal is to bridge the gap between simulation and deployment by using these stress profiles as diagnostic signals that anticipate and preempt real-world

---

### Author Response · Authors · 2025-11-30
**Rebuttal Summary**

We thank the reviewers for their constructive feedback. During the rebuttal period, we engaged in discussion regarding the validity of our "Factor Isolation" methodology, the utility of simulation-based stress tests, and the concrete architectural solutions our benchmark reveals.

We have addressed these concerns through three major empirical additions (Control-Normalized Analysis, Heuristic Validation, Ablation Studies) and a comprehensive manuscript revision.

1. Resolving the "Factor Isolation" & "Circularity" Concern (Reviewers 1, 3) The most critical critique was whether our tasks truly isolate specific capabilities or are merely "different flavors of pick-and-place." We addressed this with three distinct lines of evidence:

    - Structural Constraints: We clarified that we do not merely assume control is trivial in Perception tasks; we mechanically remove the failure conditions. By changing success criteria from Geometric Precision (Control) to Binary Contact (Perception), we physically eliminate grasp instability and trajectory jitter as failure modes.

    - Control-Normalized Analysis (New Appendix A.5): To mathematically prove success is not "driven by control," we established a baseline of the agent's motor proficiency on primitives (e.g., "Pick Cube": 96% success). We then normalized complex task scores against this baseline. The result—performance on Hard Spatial tasks drops to 0.00% normalized —proves the bottleneck is strictly the added reasoning complexity, not the Action Head.

    - Differential Diagnosis: We demonstrated that model rankings reverse across axes. Diffusion Policy dominates Control Stability (0.99) but fails Perception Stress (0%). GROOT has lower stability but succeeds at Perception. This ranking flip proves orthogonality.

2. Identifying Root Causes & Concrete Solutions (Reviewer 1) Reviewer 1 noted that while NEBULA identifies problems, it should also point toward solutions. We clarified that NEBULA moves the field from observation ("The agent failed") to Root-Cause Diagnosis:

    - The "Action Head" Bottleneck: By decoupling the VLM backbone from the execution layer (Section 5.2), we proved that standalone VLMs generate valid plans with 85-100% success , yet integrated execution fails (0-75%). This identifies the "Semantic-Execution Gap" as the primary failure mode, instructing the community to stop optimizing backbones and start optimizing grounding interfaces.

    - The "Real-Time Barrier" Solution: We identified that monolithic VLAs (e.g., SpatialVLA) fail dynamic tasks due to low inference frequency (<2Hz). In contrast, GR00T’s dual-system architecture (separating reasoning from a 17Hz diffusion spinal cord) successfully bridges this gap. NEBULA thus empirically validates Hierarchical/Dual-System architectures as the necessary solution for generalist agents.

3. Justifying Simulation-Based Stress Tests (Reviewer 1, 4) We addressed concerns regarding the sim-to-real gap by expanding the stress suite to 12 categories, including Poisson-Gaussian Noise and Command Packet Loss  to explicitly model hardware-level degradation. We frame NEBULA as a scalable "Gatekeeper", which provides a rigorous pre-deployment filter that identifies failure modes (e.g., latency sensitivity) often impossible to test consistently on physical hardware.

4. Refining Terminology (Reviewer 4) We accepted the critique regarding "overselling." We conducted a rigorous editorial pass, replacing "Ecosystem" with "Unified Framework" and removing "Novel" from "Dual-Axis" to ensure the contributions speak for themselves.

NEBULA advances the field by shifting evaluation from opaque success metrics to interpretable, root-cause diagnosis. By empirically validating the "Semantic-Execution Gap" and the necessity of Dual-System architectures, we offer a concrete roadmap for building the next generation of robust generalist agents. We hope the rigorous Control-Normalized Analysis  and our expanded stress testing suite have fully resolved the reviewers' concerns regarding factor isolation and realism. Given these extensive empirical validations and the urgent need for standardized diagnostics in VLA research, we respectfully request a positive re-evaluation of this work.

---

### Meta-Review · Area_Chair_c7as · 2026-01-07

**Summary:**

NEBULA is proposed as an evaluation-first ecosystem for VLA single-arm manipulation, aiming to move beyond coarse end-task success toward interpretable, skill-level diagnostics and robustness measurement. A dualaxis protocol is introduced: capability tests to probe specific skills (eg. spatial reasoning, grasp synthesis) and stress tests to vary conditions and expose reliability failure cliffs. To reduce benchmark and data fragmentation, a standardized api and data format are provided to unify suites (such as- ManiSkill, LeRobot), alongside an aggregated dataset spanning real demonstrations, simulator trajectories, & world-model aug. data for cross-datasets training and comparison.

In the original round of reviews the committee acknowledged the motivation, the benchmark unification effort, and clear presentation (R#iywu,krug, pvpg, zcC7). The same reviews raised concerns about unclear overlap between capability vs stress axes and some metric redundancy (R# iywu, krug, PVpg); whether the six dimensions are meaningfully isolatable or distinct (iywu, krug, pvpg); under-specified evaluation and fine-tuning protocol details (pvpg); limited discriminative power or inconsistent difficulty ordering for some dimensions (krug); and simulation-only stress testing leaving real-world validity uncertain (iywu and zcc7). Concerns were also raised that the diagnostic framing still relies largely on success-based signals and that the benchmark largely surfaces known failure modes without concrete remedies (pvpg,iyWu).

The area chair acknowledges the detailed rebuttal and post-rebuttal summary. The rebuttal and revised draft help with several clarity and completeness items raised in the initial reviews, including tighter definitions of the capability vs stress axes, more explicit protocol specifics (task and episode counts, episode length, pressure-level parameterization), and additional validation analyses that better motivate the intended evaluation design. At the same time, some central points remain only partially addressed, including the degree to which the proposed axes are cleanly separable beyond targeted checks, the reliance on simulation-only stress testing for robustness claims, and the fact that the core diagnostic signal remains largely success-based even if more structured.

Considering the above, the area chair concurs with the consensus reflected in the detailed original reviews.

**Reviewer Concerns:**

The rebuttal and revised draft help with several clarity and completeness items raised in the initial reviews, including tighter definitions of the capability vs stress axes, more explicit protocol specifics (task and episode counts, episode length, pressure-level parameterization), and additional validation analyses that better motivate the intended evaluation design. At the same time, some central points remain only partially addressed, including the degree to which the proposed axes are cleanly separable beyond targeted checks, the reliance on simulation-only stress testing for robustness claims, and the fact that the core diagnostic signal remains largely success-based even if more structured.

**Reviewer Scores:**

While it isn’t possible to predict how reviewers would have responded if ICLR had a full discussion period (cut short on 28th Nov), it seems unlikely the scores would have tipped over to clear acceptance or consensus of acceptance.

Some positive movement may have occurred based on added isolation analyses (iywu), if recalibration and metric revisions are convincing (krug), improved protocol clarity (pvpg), and writing and framing revisions (zcc7).

---

### Decision · Program_Chairs · 2026-01-26

Reject